# Investigation of Ultrasonic Treatment on Physicochemical, Structural and Morphological Properties of Sodium Alginate/AgNPs/Apple Polyphenol Films and Its Preservation Effect on Strawberry

**DOI:** 10.3390/polym12092096

**Published:** 2020-09-15

**Authors:** Wenting Lan, Siying Li, Shiti Shama, Yuqing Zhao, Dur E. Sameen, Li He, Yaowen Liu

**Affiliations:** 1College of Food Science, Sichuan Agricultural University, Yaan 625014, China; WTLan3253@163.com (W.L.); lsy237988935@163.com (S.L.); shamashiti@126.com (S.S.); mandozyq@163.com (Y.Z.); sameen0388@gmail.com (D.E.S.); 2School of Materials Science and Engineering, Southwest Jiaotong University, Chengdu 610031, China

**Keywords:** sodium alginate, apple polyphenol, strawberry

## Abstract

An antibacterial and anti-oxidation composite film was prepared by a casting method using sodium alginate (SA) and apple polyphenols (APPs) as the base material and glycerol as the plasticizer. Silver nanoparticles (AgNPs) were deposited by ultrasonic-assisted electrospray method. The degree of influence of the addition ratio of SA and AgNPs and different ultrasonic time on the mechanical properties, barrier properties, optical properties, and hydrophilicity of the composite film was explored. The composite films were characterized by Fourier transform infrared spectroscopy (FTIR), X-ray diffraction (XRD) and scanning electron microscopy (SEM). The results showed that the SA: AgNPs ratio of 7:3 and the ultrasonic time for 30 min have the best comprehensive performance, and SA/AgNPs/APP films showed the lowest water vapor permeability value of 0.75 × 10^−11^ g/m·s·Pa. The composite film has good strength and softness, with tensile strength (TS) and elongation at break (E) at 23.94 MPa and 29.18%, respectively. SEM images showed that the surface of the composite film was smooth and the AgNPs’ distribution was uniform. The composite film showed broad antibacterial activity, and the antibacterial activity of *Escherichia coli* (92.01%) was higher than that of *Staphylococcus aureus* (91.26%). However, due to the addition of APP, its antioxidant activity can reach 98.39%, which has a synergistic effect on antibacterial activity. For strawberry as a model, the results showed that this composite film can prolong the shelf life of strawberries for about 8 days at 4 °C, effectively maintaining their storage quality. Compared with the commonly used PE(Polyethylene film) film on the market, it has a greater fresh-keeping effect and can be used as an active food packaging material.

## 1. Introduction

In recent years, the research and development of new bio-based materials with biocompatibility and biodegradability to replace petroleum-based plastics needs to be solved urgently [1]. Packaging materials that are bio-based, that is, made of proteins, polysaccharides, and lipids, are non-toxic and degradable. They provide an excellent barrier, which can effectively protect food, extend its shelf life, and reduce the environmental impact of traditional packaging materials [2]. Linear polysaccharide sodium alginate (SA) extracted from brown algae is one of the most widely used biodegradable polymers; it is composed of β-d-mannuronic acid and α-l-glucuronic acid [3]. SA is widely used in medicine, food, environment, and bioengineering for its excellent properties, such as non-toxicity, wide sources, pH sensitivity, good biocompatibility, and film-forming ability. Thus, SA is an ideal base material for use in a new type of active packaging material [4]. However, traditional SA packaging films without antibacterial or antioxidant activity cannot meet the needs of active packaging. Therefore, it is necessary to introduce functional materials into the SA to provide antibacterial and antioxidant activities [5]. In order to add antioxidant activity, apple polyphenol (APP) has been applied to food preservation as a new type of biological preservative. APP has strong antioxidant activity, and has physiological effects such as lowering blood sugar, reducing fat, anti-tumor, anti-inflammatory, etc., and is used in medical research [5]. According to this, APP has been added into the active packaging materials in recent years to prepare antibacterial and antioxidant active packaging with good results. Sun et al. [6] mixed APP into chitosan to develop a new functional film. The results showed that adding APP to chitosan considerably improved the physical properties of the film and increased the antioxidant and antibacterial activities. Moreover, the addition of APP can improve the tensile strength, elongation at break and elastic modulus of the film [7]. There is no study on the preparation of active film by adding APP into SA. Yuan et al. proposed that with the increase in tea polyphenol content, the tensile strength, fracture strain and water vapor transmission rate of the film increase. Moreover, tea polyphenols can enhance its inflammatory properties and promote wound healing. The composite film with polyphenols can be used not only as food packaging, but also as wound dressing [8]. The above research fully shows that adding APP into SA to prepare active packaging materials and its application in food packaging has research significance.

It is well known that functional nanometals have been widely used as antimicrobial agents. Among the metal nanoparticles, silver nanoparticles (AgNPs) have good conductivity, chemical stability, catalysis, and antibacterial properties, which have been the focus of many studies in recent years [9,10]. AgNPs showed significant spectral antibacterial effect on Gram-negative and Gram-positive bacteria and it is engineered to be at least 100 nm ultrafine particles, the size of which is in nanometer (nm) [11]. Based on this, a large number of research studies have applied AgNPs to anticancer research, natural and synthetic textiles [12]. It is generally believed that the inhibitory effect of silver on bacteria is the reaction of silver with thiol group in protein, thus inducing the inactivation of bacterial protein. AgNPs adhere to the surface of bacteria, form pits and change the permeability of the membrane, causing content to flow out, or penetrate into bacteria, damaging its DNA [13]. However, the application of AgNPs is also often inhibited. First, because of its high surface energy, it is easy to fuse and aggregate, resulting in difficulties in processing, storage, and application. Secondly, the aggregation of nano-silver will reduce its antibacterial activity. Moreover, its particles are small in diameter and migrate easily, posing a threat to life and the environment [14]. Encapsulating AgNPs in the system can provide an effective way to prevent migration and aggregation, and SA is an excellent bio-based substrate for encapsulation. The SA film is dense and has a fast cross-linking reaction with Ca^2+^, which can effectively lock the particles, and it is feasible to apply it to nano-silver packaging. Zhao et al. [15] used SA as a stabilizer and glucose as a stabilizer to prepare antibacterial SA fiber containing AgNPs by electrospinning. The results of antibacterial experiments against *S. aureus* and *E. coli* showed that the fiber had strong antibacterial activity. Abou-Okeil et al. mixed sulfadiazine with AgNPs into hyaluronic acid/SA based films and cross-linked them with Ca^2+^, Zn^2+^ or Cu^2+^ metal cations. The antibacterial and physical mechanical properties of the cross-linked film show that the film is suitable for local bioactive wound dressing [16]. The poly(vinyl alcohol)/SA/AgNPs nanocomposites prepared by Narayanan et al. [17] showed high bactericidal activity and thermal stability, and well co-encapsulated AgNPs by SA and polyvinyl alcohol. This type of composite material offers a clear improvement in mechanical properties and water swelling ability, and it has a high and lasting antibacterial effect. The aforementioned research shows that SA can effectively embed or co-embed AgNPs with good experimental effect. The concentration of nanoparticles is an important parameter, which will affect the performance of the final machined surface [18]. The effects of nano-silver added to SA and molecular characterization need further study.

Besides, nanoparticles are close to each other in Brownian motion. When the attractive energy is greater than the repulsive energy, particle aggregation will occur. It is necessary to change the interface area between particles by physical or chemical methods to disperse nanoparticles. Among them, ultrasonic treatment will lead to high pressure, which will cause high stress, and then destroy the binding energy of nanoparticles [19]. The use of ultrasound for dispersion is also considered to be an effective way to destroy aggregates into smaller particles without the need for chemical dispersants [20]. The results show that the appropriate ultrasonic treatment time (30 min) can improve the performance of the film and effectively disperse graphene oxide and zinc oxide nanoparticles [21]. Ultrasonic treatment can also effectively reduce the size of biopolymer and form a uniform film, which has a positive impact on the mechanical properties and barrier properties of the film [22]. The dispersion of AgNPs was treated by an ultrasonic wave, which provided the possibility to solve the problem of AgNPs aggregation and improve the performance of the composite film.

In this study, SA/AgNPs/APP anti-bacterial and anti-oxidation bioactive films were prepared by using SA as the biological substrate and glycerol as a good plasticizer. Ultrasonic waves were introduced to solve the problem of AgNPs aggregation and improve the performance of the composite film. The effects of the ratio of SA to AgNPs and the duration of ultrasonic treatment on the properties of the film were studied, and the intermolecular forces were analyzed. At present, there is no study on the blending of SA, AgNPs, and APP to prepare a bioactive film, and there is no systematic study on the influence of ultrasonic time on the dispersion of AgNPs. The antibacterial efficiency of the film against Gram-negative representative bacteria (*E. coli*), and Gram-positive bacteria (*S. aureus*) was tested, and strawberry was used as the model fruit to test the effect of the film as food packaging.

## 2. Materials and Methods

### 2.1. Materials

SA and glycerol were purchased from Chengdu Kelong Reagent Co. (Chengdu, China). APP (polyphenol content 50%) was purchased from Xi’an Prius Biological Engineering Co., Ltd. (Xi’an, China). AgNPs (20 nm) were purchased from Shanghai Chaowei Nano Technology Co., Ltd. (Shanghai, China).

### 2.2. Preparation of the Films

The SA/AgNPs/APP composite film was prepared by a casting method. The solution was made of APP (1%), glycerol (2%) and different mass ratios of SA:AgNPs (total mass 2%), i.e., 10:0, 9:1, 8:2, 7:3, 6:4. First, different amounts of SA were dissolved in distilled water to get an SA aqueous solution. APP powder and glycerol were added and mixed using a booster stirrer (JJ-1, Changzhou Jintan Sanhe Instrument Co., Ltd., Changzhou, China) to fully dissolve and mix, thus obtaining a precursor solution. The solution was cast on a glass plate and dried in a digital display air drying oven (RZ101-1, Suzhou Runze Oven Manufacturing Co., Ltd., Suzhou, China) at 50 °C for 6 h. After obtaining the SA/APP films, the films were fixed as the collector for electrospray. AgNPs powders were added to 5 mL of absolute ethanol and ultrasonically dispersed using a XH-2008D ultrasonic instrument (Xianghu development Co., Ltd., Beijing, China) equipped with a reactor with a thermostatic water bath (temperature accuracy of ±1 °C), a mechanical stirrer and a microtip probe (diameter of 8 mm) for 0, 10, 20, 30, and 40 min (frequency: 40 kHz, power: 50 W). Subsequently, the suspension was filtered through a 5 mL syringe with a single needle configuration (200 μm outside diameter and 100 μm inside diameter) and was continuously pushed by a syringe pump (Zhejiang University Medical Instrument, Hangzhou, China) at 0.20 mL/h, using a electrospray apparatus equipped with 10 kV power supply (Tianjing High-Voltage Power Supply Co., Tianjing, China), as reported previously [23]. Spraying distances of 5 cm were set between the syringe nozzle and the SA/APP films. Finally, the composite film was soaked to a 2% calcium chloride aqueous solution for cross-linking (2 min), and then air-dried to obtain a SA/AgNPs/APP composite film.

### 2.3. Film Thickness and Color

The color of the film was measured on a white paper by a hand-held color difference meter (HYCS200, Anhui Huabiao Testing Instrument Co., Ltd., Hefei, China). The thickness of the film was measured by a digital thickness gauge (Insize, Loganville, GA, USA) with an accuracy of 0.01 mm. Five random points were measured on each film, and the mean value was calculated prior to determining the mechanical and barrier properties [3].

### 2.4. Mechanical Strength of the Film

Tensile strength (TS) and elongation at break (E) of the films were determined by a universal material testing machine (Ps-281, Suzhou Wanyi Experimental Instrument Co., Ltd., Suzhou, China), according to ASTM-D D412-98a [24].

### 2.5. Water Vapor Permeability of Films

Water vapor permeability (WVP, g/m·s·Pa) of the film was determined gravimetrically according to the method of ASTM E96-80 [25] with some modifications. Briefly, the complete and non-porous film was tightly sealed in the mouth of a glass cup containing filled with anhydrous calcium chloride (0% RH). The cup was then placed in a desiccator with silica gel at 20°C (1.5% RH, 28.044 Pa water vapor pressure). The weight change of the cup was recorded at 24 h intervals for 7 d, and the WVP was calculated using the following formula: (1)$$WVP=\frac{G\times L}{T\times S\times \Delta P}$$
where *G* is the constant time weight change (g), *L* is the thickness of the film (m), *t* is the time (s), *S* is the cup area (m^2^), and Δ*P* is the water vapor pressure difference (Pa).

### 2.6. Contact Angle of Water and Swelling Ratio

The contact angle of water was measured by optical contact angle measuring instrument (Theta Flex, Biolin Technology Co., Ltd., Gothenburg, Sweden, mN/m) at room temperature (25 °C) using the static drop method. The dried film sample (5 × 5 cm) was immersed in 50 mL distilled water and kept at room temperature for 24 h, and then samples were weighed before and after immersion [21]. The swelling ratio (SR) of the film was calculated using the following formula: (2)$$SR\left(\%\right)=\frac{W_{t}-W_{0}}{W_{0}}\times 100\%$$
where *Wt* is the film mass at t moment (g), and *W*_0_ represents the initial mass (g).

### 2.7. DPPH Free Radical Scavenging Activity and Antibacterial Property

The method of Riaz et al. [9] was used to measure the scavenging rate of DPPH to free radicals, and micro-modification was performed to evaluate its antioxidant activity. The DPPH measurement solution was prepared by mixing 9 mL of the thin-film extraction solution with 3 mL of a methanol DPPH solution (10^−3^ mol/L). After shaking in the spiral mixer (XH-D, Shanghai Fengchu Industrial Co., Ltd., Shanghai, China) for 1 min, it was placed in the dark for 30 min to initialize. The UV absorption of the DPPH test solution at 517 nm was measured. DPPH scavenging activity was calculated as follows:(3)$$DPPH\;scavenging\;activity\left(\%\right)=\frac{A_{D}-A_{S}}{A_{D}}\times 100\%$$
where *A_D_* is the absorbance value at 517 nm of the DPPH methanol solution and *As* is the absorbance value at 517 nm of the DPPH assay solution.

For the antibacterial test, the Gram-positive bacterium *S.*
*aureus* PTCC 1112 was first incubated at 37 °C for two passages. 1 mL of the bacterial solution was placed in a conical flask, and 1 g of composite film and 50 mL NA culture medium (1 mL of broth and 499 mL of normal saline) were added. The conical flask was placed in a 37 °C constant temperature shaking incubator and incubated for 24–48 h. Then, 1 mL of the dilution was spread on a clean nutrient agar plate and incubated at 37 °C for 24–48 h; afterward, the bacterial colonies were counted and recorded. The antibacterial test method of *E. coli* (PTCC 1270) was the same as above [4].

### 2.8. Thermal Properties

The thermal properties of SA, SA/APP, SA/AgNPs/APP films (5 mg) were determined by differential scanning calorimetry (DSC) (Q200M, Xiamen Chongda Intelligent Technology Co., Ltd., Xiamen, China). Each sample was heated from 25 °C to 350 °C in an aluminum pan at a rate of 20 °C/min under a nitrogen atmosphere [26].

### 2.9. Characterization of the Films

FTIR spectroscopy (Nicolet iS10, Waltham, MA, USA) was used to analyze SA, APP, SA/AgNPs/APP FTIR spectra. The wave number range was 4000–600 cm^−1^, and the resolution was 2 cm^−1^. The changes of peak intensity and wave number provided the information of structural chemistry. X-ray diffraction analysis (XRD, TF5000, Dandong Tongda Technology Co., Ltd., Dandong, China) of crystals of SA, AgNPs, APP, and composite film was carried out. The surface microstructure of the thin film was observed by scanning electron microscope (KYEM6200, Shanghai Weihan Optoelectronics Technology Co., Ltd., Shanghai, China) for gold spraying. Observation of the morphology and size of pure AgNPs powder with transmission electron microscope (HT7800, Chengdu Xiye Trading Co., Ltd., Chengdu, China) was perfomed [4].

### 2.10. Strawberry Fresh-Keeping Performance Test

In the experiment, the 8-point, uniformly-colored strawberry (Fragaria × ananassa Duch. cv. Benihoppe), and the post-harvest strawberries were pre-cooled and randomly grouped. After weighing and recording all strawberries, all strawberries were directly wrapped with SA/AgNPs/APP, SA/APP and SA films which were made before by electrospray. The strawberries without film were used as air control group and commercially available PE film wrapped strawberries as a PE control group. The strawberries of all groups were stored in the refrigerator with 75% humidity and 4 °C temperature. We took out the strawberries every 2 days, removed the external film for weighing, photographing, sensory evaluation and hardness measurement, and then ground the strawberries with a mortar to test other indicators over, a total of 12 days; each test was conducted 5 times.

### 2.11. Weightlessness, Sensory, and Decay

After taking out the strawberries, we directly tore off the film wrapped on the surface for weighing. The weightlessness was tested with an analytical balance (FA124, Shanghai Meiyingpu Instrument Manufacturing Co., Ltd., Shanghai, China), which was estimated as the percentage of the initial weight loss:(4)$$Weightlessness\;\left(\%\right)=\frac{Initial\;weight-final\;weight}{Initial\;weight}\times 100\%$$

According to Table 1, the decay rate of strawberry is evaluated and then calculated by the following formula [27]:(5)$$\sum^{\;}\frac{Decay\;level\times Quantity}{Highest\;decay\;level\times Total\;fruit\;quantity}$$

A total of 20 consumers were recruited to study the effects of strawberry external sensory during the strawberry preservation period. In each session, strawberry samples were provided to each consumer in a random and independent manner, and each consumer received a table containing a nine-point hedonic scale (1 = very much dislike; 9 = very much like) to evaluate the sample [28].

### 2.12. Firmness, pH, Soluble Solids Test, and Titratable Acid

A texture analyzer (GY-3, Hangzhou Zhuoqi Electronics Co., Ltd., Hangzhou, China) equipped with a P5 probe was used to measure the hardness of the strawberry fruit, and the results were expressed in Newton (N) peak force. Soluble solids and pH were measured with a hand-held sugar meter (WZS, Hangzhou Junsheng Scientific Equipment Co., Ltd., Hangzhou, China) and pH meter (PHS-3C, Tianjin Shunuo Instrument Technology Co., Ltd., Tianjin, China). The titratable acid was measured by titrating 20 mL of 10% strawberry juice sample through a basic burette using 0.01 mol/L NaOH solution [27].

### 2.13. Vitamin C (V_C_)

One hundred mL of 50 g/L trichloroacetic acid was used to extract 10 g of strawberry homogenate for 10 min, and then it was centrifuged at 3000 rad/min for 10 min. One milliliter of the supernatant was removed. To the supernatant, 1 mL of trichloroacetic acid (50 g/L), 1 mL of phenanthroline/ethanol solution (5 g/L) and 0.5 mL of FeCl_3_/ethanol solution (0.3 g/L) were added. After fully mixed, the absorbance was measured at 523 nm [28].

### 2.14. Statistical Analysis

All data were repeated 5 times, and using the statistical data of SPSS 24 software, the significant difference between the means was determined by Duncan’s multiple range test at a significance level of 0.05.

## 3. Results

This section may be divided by subheadings. It should provide a concise and precise description of the experimental results, their interpretation as well as the experimental conclusions that can be drawn.

### 3.1. Optical Properties of the Film

With the increase in the amount of AgNPs added, the color intensity of the film also increased, which is verified in the instrument measurement data of optical properties (Figure 1a–d). When the ratio of SA and AgNPs was 9:1, 8:2, 7:3 and 6:4, the brightness (L*-value) of the film decreased significantly (*p < 0.05*) from 60.15 ± 1.37 to 51.32 ± 0.99, 45.39 ± 3.09, 45 ± 0.93, and 40.58 ± 0.87, respectively, which means less light reflected from the film surface. With the increase in AgNPs concentration in the film, the color coordinates (a*-value and b*-value) decreased significantly, and the trend of green and blue changed. Specifically, it had the most obvious effect on the a*-value, with a maximum reduction of 98.62% (6:4, no ultrasound). The high value of color difference (ΔE) also shows that it has a significant effect on the appearance of the film, while the nanocomposite film is blackened due to the plasmon effect of nanoparticles. This finding is similar to the study of Saedi et al. [29]. As shown in Figure 1e, the SA/APP (10:0) composite film had a homogeneous yellow appearance, but the film became dark brown when AgNPs was incorporated. This is consistent with the previous results, and polyphenols lead to the decrease in brightness and increase in opacity of SA composite film [30]. Compared with the transparent and yellowish control group (10:0), the color change of the composite film can be attributed to the original color of AgNPs. In addition, the brightness, the red and yellow values of the composite film depend on the AgNPs content. Compared with different ultrasound times, the value of L*, a*, and b* of the composite film increased first and then decreased after ultrasound treatment, and 30 min is the peak and turning point. The dispersion of AgNPs in the SA/APP matrix promoted by ultrasound was uniform, and the nanoparticles were well separated from each other [31]. First, ultrasound resulted in more uniform color, while longer ultrasound time led to agglomeration of nanoparticles, leading to color change. Borah et al. [32] showed the same trend. The concentration of AgNPs in the final films is the main reason, and the difference of biopolymer size caused by ultrasonic time is the second reason for the difference in the color of the films.

### 3.2. Mechanical Properties Analysis of Film

Table 2 shows the tensile strength (TS) and elongation at break (E) of the SA/AgNPs/APP films. It was observed that the ratio of SA and AgNPs was 10:0 with no ultrasound and the TS was lower than other films with ultrasound treatment. The presence of AgNPs caused a decrease in E values; SA/AgNPs composite film showed the lower mechanical properties because of its high specific strength (*p* < 0.05). AgNPs and APPs are easy to be inserted into the network of SA, resulting in covalent cross-linking, which affects the mechanical properties of the SA film [30].

When a certain amount of AgNPs was added, the TS value of the blend film decreased significantly, which may be due to the heterogeneity of the material structure network caused by the agglomeration of nanoparticles. The agglomeration phenomenon may be caused by the decrease in the distance between Ag particles, the decrease in aggregation factor (van der Waals force attraction) and dispersion factor (electrostatic or spatial repulsion), which leads to irreversible agglomeration and the decrease in strength [31]. In addition, it was closely related to the distribution and density of intermolecular and intramolecular interactions between polymer chains in the film matrix [33]. The composite film needs to be cross-linked with calcium chloride, and the TS of SA after cross-linking is large. As a result of the addition of AgNPs, the cross-linking degree of SA matrix and calcium chloride increased, which led to the simultaneous decrease in TS. Higher mechanical properties can be explained by the formation of hydrogen bonds between the SA and APP, which make them more compatible with the interface filler matrix. However, the addition of AgNPs leads to the decrease in hydrogen bond interaction between SA and APP, the separation of film liquid phase, and the damage of compatibility between SA and APPs [34]. This conclusion is similar to the study of Kim et al. [35]. With the increase in the proportion of AgNPs, the TS of tragacanth/hydroxypropyl methylcellulose/beeswax edible film decreased significantly, which was closely related to the distribution and density of intermolecular and intramolecular interactions between polymer chains in the film matrix [33]. The composite film needs to be cross-linked with calcium chloride, and the TS of SA after cross-linking is large. Due to the addition of AgNPs, the cross-linking degree of the SA matrix and calcium chloride increased, which led to the simultaneous decrease in TS.

At the same time, E values were increased with the addition of AgNPs, and reached the highest value of 29.18 ± 1.59% at a ratio of 7:3. According to the influence of different ultrasound times on the SA/AgNPs/APP composite film, it was found that the E values of the composite film increased first and then decreased with the increase in ultrasound time from 10 min to 40 min. Clearly, 30 min of ultrasound is the best amount of time for enhancing the uniformity of nano-silver in SA/AgNPs/APP composite film, enhancing the rigidity and ductility of the composite film. It should be noted that the measured values depend on the sample moisture content, cross-linking density, and other properties of the matrix, but the trend here is still valid [31]. When the ultrasonic treatment time reached the threshold value (40 min), the dense and uniform network structure of the composite film is disturbed, resulting in the aggregation of nanoparticles, which has a negative effect on the mechanical properties of the composite film [23]. The TS and E values of SA/chitosan/AgNPs composite film were 23.80 MPa and 16.55%, respectively. Compared with SA/APP/AgNPs composite film (TS = 23.94 MPa, E = 29.18%), its mechanical properties were relatively low, which fully demonstrated that APP had a positive effect on the mechanical properties of the composite film. It also shows that the composite film has good mechanical properties and the composite film has a bright future [36]. SA/guava leaf extract composite film was prepared by Luo et al. Due to the hydrogen interaction between SA and guava leaf extract, the structure of the composite film was more compact, which improved the TS of the composite film. However, the excessive ratio of extract may lead to uneven dispersion in the mixture, resulting in the decrease in TS [37].

### 3.3. WVP Analysis of Film

Figure 2a shows WVP values of the SA/APP and SA/AgNP/APP films. Barrier properties of the composite films without the AgNPs (10:0) were also included for comparative purposes. Compared with the control group (10:0, 1.6 × 10^−11^ g/m·s·Pa), the addition of AgNPs significantly reduced the barrier property of the film. The ratio of 9:1 and 8:2 caused a significant (*p* < 0.05) increase in the WVP values, which were 2.25 × 10^−11^ g/m·s·Pa and 2.17 × 10^−11^ g/m·s·Pa, respectively. This increase may be due to the reduced intrachain interaction of SA and the formation of voids in the film structure. This result is similar to that of Jafari et al. [38]. Adding AgNPs to the chitosan matrix increases the WVP value from 8.71 × 10^−10^ g/m·s·Pa to 9.1 × 10^−10^ g/m·s·Pa. However, due to the substrate, the barrier properties of the SA/AgNPs/APP composite film are significantly better than the SA/APP composite film. It was clearly observed that continued addition of more AgNPs slightly reduced the WVP values, a fact which can be attributed to the increased hydrophobicity of the film. WVP of SA/AgNPs/APP films with AgNPs (7:3) significantly declined (*p* < 0.05) to 1.89 × 10^−11^ g/m·s·Pa. AgNPs could insert into the network structure of SA to form a denser system, which led to a reduction in WVP values [30]. Similar results were also observed in a study by Bang et al. [39], where in the WVP value of gelatin the composite film decreased from 1.53 × 10^−9^ g/m·s·Pa to 1.47 × 10^−9^ g/m·s·Pa due to the addition of AgNPs. It is believed that nanoparticles can reduce the permeability of water vapor by increasing the crystallinity of biopolymers or reducing the free hydrophilic groups in alkyl groups, such as -OH, thereby increasing the diffusion channel of water vapor [39]. WVP of films ranged from 2.25 × 10^−11^ g/m·s·Pa to 0.75 × 10^−11^ g/m·s·Pa, which was significantly affected by the different ultrasound treatments, and the lowest WVP value is displayed in 7:3–30 min (0.75 × 10^−11^ g/m·s·Pa). This effect can promote the cross-linking between SA and APP molecules and increase the degree of cross-linking between SA and CaCl_2_ to form a thinner and more compact structure. A study by Cruz-Diaz et al. [40] showed the same results. The application of ultrasound can also make AgNPs better distribute in the film, and WVP is also generally lower in denser film networks. It can be seen from the WVP value (1.30 × 10^−9^ g/m·s·Pa) of SA/APP composite film that APP contributes to the low WVP of composite film, but its WVP is mainly affected by the type of substrate and the preparation method of composite film and the degree of cross-linking [41]. SA is the only natural polysaccharide containing a large number of hydrophilic carboxyl groups. Its unique property is that it reacts with multivalent metal cations (especially calcium ions) to produce insoluble polymers. Therefore, the treatment of SA with CaCl_2_ solution can effectively reduce its hydrophilicity [42]. Moreover, the strong interface interaction between polyphenols and biomacromolecules can form a more compact composite film system, thus improving the water resistance of the composite film. The intermolecular hydrogen bond interaction between APP and SA will also result in the lower WVP of the composite film, and the cross-linking between SA and CaCl_2_ is also an important reason for maintaining a high barrier [43]. Luo et al. also suggested that the strong interfacial interaction between phenolic compounds and sodium alginate may improve the moisture resistance. However, the high content of phenolic compounds may accumulate in the polymer matrix to produce voids, resulting in a higher WVP value [37]. Compared with the SA/guava leaf extract composite film (0.55–0.75 × 10^−11^ g/m·s·Pa), the WVP value of the film had no significant difference, and the water vapor barrier of the two films was better [37].

### 3.4. Water Contact Angle and Swelling Ratio Analysis of Film

The water contact angles of the SA/AgNPs/APP composite films are shown in Figure 2b. The SA/APP (10:0) composite film showed a superhydrophilic surface, while its casted film had a contact angle of 56.6°. The 7:3 film had a contact angle of 61.2°, which was more hydrophobic than the 9:1 casted film (25.6°). Interestingly, the 8:2 film had a hydrophilic surface with a water contact angle of 46.3°, while the 6:4 film showed a hydrophilic surface at 57°. The increase in the water contact angle of the AgNPs incorporated nanocomposite films may be due to the hydrophobicity of metallic silver [39]. The contact angle shows the hydrophilicity (<90°) and hydrophobicity (>90°). Clearly, SA/AgNPs/APP composite films are hydrophilic films. However, the lower hydrophilicity of SA/APP is due to the cross-linking of CaCl_2_ and the interaction between APP and SA, which leads to the less free -OH on the surface of the film, resulting in the lower hydrophilicity of the surface. It is well known that adding surface roughness will add to the wettability caused by the chemistry of the surface [27].

The effect of the ratio of SA with AgNPs and ultrasonic treatment time in the swelling-ratio behavior properties of SA/AgNPs/APP films was evaluated, and the results are shown in Figure 2c. SRs of SA/AgNPs/APP films were 60.90% (10:0), 45.85% (9:1), 38.29% (8:2) and 36.23% (6:4), respectively, showing the high water hydrophilic characteristic. The higher the concentration of SA, the higher the swelling degree of the film preparation, which can also be observed in the SA/AgNPs/APP film [44]. After added AgNPs, the minimum SR of the SA/AgNPs/APP films reached 28.37% (decreased by 32.63%) in 7:3. The high values of SR of the SA/APP (10:0) films may be attributed to the strong hydrophilicity of SA and APP that can easily interact with water molecules [6]. The decrease in SR indicated a water hydrophobia of the SA/APP films that was enhanced for the hydrophobic characterization of AgNPs. These differences in the SR changes could be due to the SA/APP content used in each formulation. First, increasing the SA/APP concentration results in a higher number of reactive sites to interact with water, thus increasing the water absorption. On the contrary, it shows why the concentration of AgNPs increases while the SR value decreases. The number of -OH in SA/APP film decreased after incorporation of AgNPs, thus it could combine less water [35]. In addition, this may be due to the interaction between the SA polymer chain and AgNPs, which limits the movement chain of the SA polymer [4]. It is worth noting that the ultrasonic treatment can significantly reduce the SR of the composite film. When it is 7:3–30 min, the lowest SR is 7.52%. Due to the special behavior of the grain boundary, the hydrophobic repulsion of AgNPs is caused, while the ultrasonic wave disperses AgNPs uniformly in the matrix for a certain time, which hinders the swelling caused by water molecules. Secondly, the uniform AgNPs has a certain resistance to the invasion of water molecules, which eventually leads to the decrease in SR. However, the ultrasonic time transition leads to the aggregation of AgNPs and the formation of pores, which results in the increase in SR [20]. As for the decrease in 10:0 SR caused by ultrasound, it may be due to the more closed matrix caused by ultrasound, which makes the blend film more difficult to contact water, and the surface cross-linking degree of CaCl_2_ is high [32]. Su-Giz et al. pointed out that the SR limit of the cross-linked SA film in water is between 50% and 70%, and the SR of the SA/AgNPs/APP film is within the normal range. The prepared composite film can rapidly swell in water for 5 min, which is very useful for the release of active substances. The SA composite film is suitable for wound dressing and food packaging [45]. The swelling percentage of sodium alginate composite film in acid buffer solution is up to 95%, and it has great swelling characteristics under acidic conditions. When the composite film is applied to strawberry preservation research, the rotten juice of strawberry can stimulate the film swelling and release antibacterial and antioxidant substances, which brings advantages to the system [44].

### 3.5. FTIR and XRD Analysis of Film

The FTIR of SA, APP, SA/APP and SA/AgNPs/APP are shown in Figure 3a. In the SA spectrum, the absorption peaks of the COO^-^ group asymmetric stretching vibration and symmetric stretching vibration were observed at 1588 cm^−1^ and 1417 cm^−1^, and the vibration of -OH shows a broad peak at 3210–3520 cm^−1^, while the other band observed at 2923 cm^−1^ can refer to as the vibration band of group C-H. In addition, the SA and SA/AgNPs/APP film, including SA/APP film, show a band at 1026 cm^−1^ that corresponds to the C-O and C-O-C groups [2]. APP is a kind of biomolecule with strong antioxidant effect. It contains a lot of proanthocyanidins, flavonols, catechins, and epicatechins, which are beneficial to human health. In the APP spectrum, the absorption peak at 2922 cm^−1^ is derived from the symmetric vibration of the alkyl unit and the tensile vibration of C-H in the aromatic methoxy group [9]. It is possible to observe a peak in the 1023 cm^−1^ regions, which is attributed to the tensile vibration of C-O in the C-O-C group. The broad peak of polyphenols near 3420 cm^−1^ corresponds to O-H stretching, which is affected by intermolecular or intramolecular hydrogen bonds [43]. The characteristic absorption peak of polyphenols at 1616 cm^−1^ and 1521 cm^−1^ is attributed to the C=C vibration of the polyphenol skeleton [30]. The absorption peak of this FTIR spectrum shows the characteristic bands of free radical groups present in flavonoids and hydroxycinnamic acid, which are composed of two main polyphenol compounds present in apples. This is similar to the conclusion of Sun et al. [6]. With the addition of polyphenols, the absorption peak of -OH gradually shifts to the number of wavenumbers (3283 cm^−1^ to 3254 cm^−1^). The above changes indicate that the addition of tea polyphenols may cause changes in certain specific groups. Therefore, polyphenols can be selected to be cross-linked with SA [30]. It was observed from the SA/APP spectrum that with the addition of APP, the broad peak at 3286 cm^−1^ was flatter, which was related to the reduction in the tensile vibration of free -OH caused by the hydrogen bond interaction between APP and SA [9]. This phenomenon is similar to that of Luo et al. The shift of the absorption peak indicates that there is an interaction between the sodium alginate and plant extract. The change of the peak position and peak intensity is caused by the hydrogen bond between polyphenols and SA. The stronger intermolecular interaction can improve the physical properties of the films [37]. As a result of the addition of AgNPs, the absorption peak of 3254 cm^−1^ shifts to 3267 cm^−1^, which is similar to that of Sharma et al. Additionally, it shows that the absorption peak of -OH moves to the long wavelength due to the formation of the bond between SA and AgNPs [36]. SA/AgNPs/APP and SA/APP composite films with different ultrasound times have similar FTIR spectra as SA, which shows that the ultrasound cannot increase or decrease the functional groups in the composite film, but the changes caused mainly break the intermolecular hydrogen bonding between SA and APP [43].

The crystal structure of the SA/AgNPs/APP composite film was investigated through XRD (Figure 3b). The diffraction patterns of SA/AgNPs/APP composite films showed the characteristic peaks of AgNPs at 2θ = 38.2° and 44.38°, corresponding to (111) and (200) crystal planes for silver particles in AgNPs. These results confirmed that nano-scale metallic AgNPs are embedded in the composite film. Moreover, four diffraction peaks at 2θ = 64.5° and 77.5° were observed. These small but significant peaks are attributed to Ag crystals since they correspond to the diffraction planes of (220) and (311) of AgNPs, respectively [46]. This is similar to the conclusion of Sharma et al. [36]. In addition, the crystalline phase of the SA film is very low, and only has a broad and non-prominent peak, which is the same as the results reported in previous studies of Shaari et al. [26]. SA/APP composite film is dominated by amorphous structure, and the peak of the crystalline phase is wide and insignificant. Once nanoparticles are added to the SA/APP film, there will be a peak. For example, in the SA/AgNPs/APP composite film, there will be a characteristic peak of AgNPs, but its strength is lower than that of pure AgNPs. This conclusion fully illustrates that AgNPs are embedded in the SA/APP matrix. The diffraction peaks of XRD increase with the increase in concentration, and the arrangement of peaks does not change with the formation of the composite film, which fully reflects the crystal structure [26]. Pure APP showed a wide peak at 2θ = 19.72°, which was similar to the XRD results of apple polysaccharides in “Qinyang” in the study of Hou et al. [47]. The results showed that the apple extract had a semi crystalline and amorphous structure, and the crystal structure might be caused by the partial arrangement of polysaccharide molecules. It can be observed that after APP and AgNPs are added to SA, their positions show an increasing trend, and this indicates that the three have intermolecular interactions. In addition, Gao et al. [48] pointed out that the ultrasonic treatment can improve or reduce the crystal surface activity of the nanoparticles, resulting in the peak value, which is related to the medium.

### 3.6. TEM and SEM Analysis of Film

As shown in Figure 4a, AgNPs are spherical, ellipsoidal, or cylindrical, and their particle size range is 23–42 nm, and this is just like the morphology and granularity of AgNPs in previous research [49]. It conforms to nanoscience’s definition of nanoparticles with a particle size of 1–100 nm. The high antibacterial activity of AgNPs is due to their large surface area, which can directly interact with microorganisms, while the antibacterial activity of AgNPs is related to particle size [49].

The SEM of the 10:0 (SA/APP) film is very smooth, continuous, and without holes, which shows that the APP was well-distributed into the SA matrix. This is similar to the study of Sun et al. [6]. It was postulated that the APP was evenly distributed in the SA film forming matrix. Dou et al. [30] pointed out that polyphenols and the matrix can form a cross-linked network, with the increase in polyphenol concentration, it shows a denser internal structure. In addition, the AgNPs were clearly observed and present uniform distribution in the matrix (red arrow). At a higher concentration of AgNPs (6:4–30 min), the rigid and rougher surfaces are more marked. It was also observed that the AgNPs started to agglomerate at a higher concentration in the SA matrix (Figure 4b). SEM confirms the relatively homogeneity of AgNPs particles in the SA matrix in 7:3–30 min, compared with other concentrations (9:1, 8:2, and 6:4) and ultrasound times (10, 20, and 40 min). Ultrasonic wave propagates in the form of sound wave in the solution, which leads to the rupture of microbubbles in the film. It can make particles uniformly distribute in solution on a macro scale, while the cavitation effect can make particles more uniform on a micro scale [23]. However, if the ultrasound time is too long, it will lead to aggregation again. This was verified in this study, similar to the Zhang et al. [20]. Notably, 7:3–30 min is the smoothest composite film with a microstructure, which also verifies the influence of different ratios and ultrasonic times on the barrier properties, mechanical properties and hydrophilicity of the composite film.

The EDS spectrum obtained under the SEM shown in Figure 5 clearly shows the main peak of Ag and the small peak of Ca^2+^, confirming the content of Ag. The peaks corresponding to the substrate elements (C, N, O) were also detected. It can be seen that the composite film is effectively cross-linked by Ca^2+^.

### 3.7. DSC Analysis of Film

DSC analysis was used to determine the thermal stability of the film. The DSC curve is shown in Figure 6. The obtained DSC results show the typical outline of SA film, as previously reported by Kim et al. [35], and the SA/APP and SA/AgNPs/APP composite films are based on SA, so the outline is similar to pure SA. The weight loss of pure SA polymer is mainly divided into two stages. The first stage is the weight loss of pure SA polymer at 200 °C, which is called the water evaporation stage. The second stage is weight loss between 250 °C and 500 °C due to depolymerization of the polysaccharide network and complex dehydration [26]. The endothermic peaks of SA, SA/APP and SA/AgNPs/APP composite films are at 123 °C, 117 °C and 111 °C, respectively, and these peaks may be due to the loss of free water. Regarding the melting temperature (Tm), the SA film has the highest melting point in the sample. With the addition of APP and AgNPs in sequence, the Tm value decreases, showing a tendency to reduce thermal stability. It can be attributed to the addition of APP and AgNPs which cause changes in the film structure [35]. The glass transition temperature (Tg) of the film also showed the same trend. The Tgs of SA and SA composite films were 108 °C, 102 °C, and 92 °C, respectively. Unlike the findings of Kim et al. [35], the Tg of SA/starch is 62–80 °C, and the reason for this result is that we have used CaCl_2_ cross-linking treatment which has a larger Tg, which increases the thermal stability of the composite film. Previous studies pointed out that changes in Tm and Tg are related to crystallinity [2]. At the same time, the thermal performance of the composite film also depends on the mobility and tightness of the polymer chains in the matrix and its crystal structure. It is worth noting that the Tg value of the mixed sample moves to a lower temperature than that of the pure SA, indicating that APP and AgNPs are well embedded in the alginate matrix. This increases the distance between macromolecules, resulting in an increase in the free volume, which enhances the mobility of molecular fragments [43]. It is pointed out that plant extracts have a plasticizing effect on the SA matrix, which can reduce the thermal stability mainly by affecting the amorphous part of polymer [2]. The exothermic peaks of SA, SA/APP and SA/AgNPs/APP composite films are at 224 °C, 235 °C, and 242 °C, respectively [26]. Over 200 °C, the polysaccharides began to decompose, including random cleavage of glycosidic bonds, evaporation and elimination of volatile products [50]. APP showed higher carbohydrate content and higher molecular weight, which may be related to the more compact and complex macromolecular structure and more stable to thermal degradation. The research of Donato et al. also showed this result [51]. On the other hand, due to the strong hydrogen bonding between alginate and AgNPs, the introduction of AgNPs slightly increased the residue and enhanced the thermal degradation temperature [52].

### 3.8. Antimicrobial Activity and Antioxidant Activity Analysis of Film

Table 3 shows the APP has antibacterial activity against *S. aureus* (27.61 ± 0.53%) and *E. coli* (27.84 ± 0.49%), and its antibacterial activity against *E. coli* is better. Previous studies have also shown this conclusion, which is due to the fact that phenols not only inhibit growth, but also have bactericidal effect. The biologically active phenolic compounds can exert the physiological changes of microbial cell films and eventually lead to bacterial death [9]. Luo et al. showed that the guava leaf extract had an antibacterial effect on *E. coli* and *S. aureus*, and the higher the concentration, the more obvious the antibacterial effect. Additionally, they suggest that different types of polyphenols have different effects on bacteria, and that polyphenol mixtures are usually much more active than any single polyphenol alone [37]. This may be due to the different sensitivity of microorganisms to different polyphenols. The sensitivity of bacteria to polyphenols depends on the species of bacteria and the structure of polyphenols [53]. Catechins have a direct antibacterial effect by destroying bacterial cell membrane, inhibiting fatty acid synthesis and inhibiting enzyme activity [54]. Plant extract is a complex mixture of many active substances acting simultaneously. Therefore, the antibacterial and antioxidant properties may play an accumulative role in its action [54]. This conclusion was also put forward in Olszewska et al.’s study [55]. In addition, biological activity may interfere with the results. Therefore, further study on the model system of plant extracts may be very helpful to study the possible synergistic or antagonistic effects. It was worth noting that the composite films functionalized by AgNPs exhibit higher antibacterial activity to both *E. coli* and *S. aureus* than the 10:0. Moreover, the results showed that the antibacterial activity of SA/AgNPs/APP films followed a AgNPs dose-dependent manner. When the concentration of AgNPs reaches 7:3, the inhibition rate of the composite film to *E. coli* and *S. aureus* can reach more than 90%, while when the concentration is 6:4, the inhibition rate to *E. coli* (92.78%) and *S. aureus* (92.48%) is the highest. Our results showed that the antimicrobial activity of SA/AgNPs/APP was stronger against *E. coli* than against *S. aureus*, consistent with the previously reported results [42]. This result is the synergistic antibacterial effect of APP and AgNPs. Every barber knows that this difference is mainly due to the cell wall structure of bacteria. Gram-positive bacteria have a thick cell wall structure with multi-layer peptidoglycan. However, Gram-negative bacteria have a complex cell wall structure with a thin peptidoglycan layer, and the outer membrane is surrounded by 23 outer membranes. The inhibition of AgNPs on bacteria is as follows: firstly, nanoparticles can combine with the surface of bacteria, change the characteristics of the membrane, make the content of bacteria flow out, and cause its death. Secondly, AgNPs can penetrate the interior of bacteria, cause DNA damage and affect the proliferation of bacteria [56]. As mentioned earlier, the bactericidal efficacy of AgNPs over a range of 5–100 nm is significantly enhanced as the size of the nanoparticles is reduced below 10 nm. When the particle size of AgNPs is 5 nm, AgNPs can kill *E.coli* rapidly in 60 min, and the bacteriostatic rate can reach 99% [57]. AgNPs has antibacterial effect on Gram-positive bacteria and Gram-negative bacteria. The difference is that Sharma et al. proposed that AgNPs have a higher antibacterial effect on Gram-positive bacteria. This is related to the positive and negative charges on the surface of AgNPs, and the negative charge has a stronger antibacterial effect on Gram-positive bacteria [37].

Table 3 shows the antioxidant activity of the SA/AgNPs/APP film, and SA shows low antioxidant activity (1.01 ± 0.53%), but its effect on antioxidant activity can be considered negligible, so it is considered as the control value. Films containing APP showed significantly higher radical scavenging activities (*p* < 0.05). The addition of 1% APP significantly increased the average DPPH radical-scavenging activity to 97.40%. In addition, the overall antioxidant capacity of the SA/AgNPs/APP film may also be the result of the synergistic action of different active substances including phenolic acids, flavonols, flavonoid-3-ol, anthocyanins, dihydrochalcone and procyanidins in the matrix rather than single compounds. Plant extracts are a complex mixture of many antioxidants that act simultaneously. Therefore, antioxidants may have cumulative effects, but synergistic and antagonistic effects are also possible. In addition, there may be interference between antioxidants and biomaterials [53]. The antioxidant activity is closely related to the purity and source of the classification, and the antioxidant activity is also closely related to its antibacterial property [58]. The application of APP in litchi preservation can significantly inhibit the browning of litchi, enhance the activity of antioxidant enzymes, and improve the shelf life of litchi [9]. Phenolic compounds with biological activity can play a role in physiological changes of microbial cell membrane and eventually lead to bacterial death. The results showed that APP had an important effect on inhibition of fruit surface microbial activity and browning. The application of the film in fruit preservation has a development prospect [6]. Therefore, it is of great significance to apply the SA/AgNPs/APP composite film containing APP in food, such as strawberry preservation and delaying strawberry aging.

### 3.9. Strawberry Fresh-Keeping Performance

#### 3.9.1. Sensory Evaluation and Sensory Evaluation Score

Table 4 shows the sensory evaluation scores of each group of films over a 12-day storage period. Figure 7 shows a picture of strawberries during storage. It can be seen that there is a downward trend in each group during the 12-day test period, especially in the air control group. The air control group had a better appearance on the second day, and on the 4th day, it began to show clearly rotten patches, until the 10th day, the large area shrank and turned black. Jiang et al. proposed that the fungal activity on strawberry fruit directly affected the shelf life of the strawberries, while the antimicrobial activity of the film led to the decrease in microbial respiratory activity and inhibition of bacterial growth, thus prolonging the shelf life of the strawberries [59]. The effect of carboxymethyl cellulose/guar gum/Ag on strawberry preservation showed that the film effectively delayed the weight loss of the strawberries. This is due to the fact that menthol has antioxidant and antibacterial effects, and Ag nanoparticles lead to slow aging and inhibit decay. The shelf life of strawberry fruit was prolonged by 8 days, which shows that the film was a promising material for strawberry packaging [60]. The difference of the PE control group was that it had a good appearance after 4 days, a small amount of plaque, soft epidermis, and a small amount of shrinkage on the 6th day, which was evaluated as 6.3 points. On the 8th day, the surface of the strawberry fruit was full of mold hyphae, which accelerated the decay and on the 10th day, the strawberries were full of mold. The reason is that PE has a high barrier, which blocks oxygen and carbon dioxide, and inhibits strawberry respiration in the early stage. However, in the later stage, it blocked a lot of water vapor produced by strawberry respiration and accumulated water in the interior, which accelerated the decay and deterioration trend of the strawberries [61]. The same conclusion can also be observed in the study of Yansu et al. The strawberry fruit coated with PE kept intact in the early stage, but the subsequent observation found that the strawberry decay was accelerated, and the degree of decay was high due to PE coating. Briefly, 4 °C was better than 25 °C in preserving strawberries [62]. Compared with the SA control group and the SA/APP control group, strawberries had complete and acceptable appearance 6 days after storage, with scores of 6.8 and 7.3, respectively. On the 8th day, there was obvious decay, and the surface of the SA control strawberry fruit also had a noticeable mold colony. However, although the strawberry fruit covered by SA/APP is rotten, it can significantly inhibit the growth of strawberry fungi, and there is no noticeable fungal colony on the surface of the strawberry fruit. This film formed by SA can seal the surface of the strawberry, which has a certain effect of inhibiting strawberry respiration [63]. SA/AgNPs/APP composite film had the best appearance phase. On the 8th day, there were a few rotten patches on the surface of the strawberry fruit covered by the composite film in the experimental group, and on the 12th day, it still had 6.5 points, which was much better than all the control groups. Compared with air control, SA/AgNPs/APP composite film can effectively block the water loss of strawberries in the refrigerator and retain the water content of the strawberries. Compared with the PE control, SA/AgNPs/APP composite film has certain water permeability, which can effectively discharge the water vapor generated by strawberry respiration and avoid strawberry decay due to water [27]. Compared with SA control, it has strong antibacterial and antioxidant properties, can effectively inhibit the microbial growth on the surface of strawberries, and prolong the shelf life of strawberry fruit. Lan et al. [27] also pointed out that the strong antioxidant ability of classification can also remove free radicals in strawberries to extend their shelf life. The research of Kanikireddy et al. showed the same trend. During storage, the weight loss rate and decay rate of unpacked strawberries were higher than those packed with carboxymethyl cellulose/guar gum/AgNPs composite film, and the water moved to the surrounding environment faster. Due to its antioxidant effect of mint leaf extract and antibacterial effect of AgNPs, the decay and ripening of the strawberries were inhibited [60]. Similarly, after the strawberries were stored at 4 °C for 3 days, mold appeared on the unpacked samples. The plyvinyl alcohol composite film added with tea polyphenols could effectively prolong the shelf life of strawberries by 3 days and effectively inhibit the biological activity of mold [27]. Due to the excellent antibacterial property of AgNPs, the preservation effect of the SA/AgNPs/APP composite film on the strawberries was better. SA has a high barrier, which can effectively inhibit the transpiration and respiration of fruits, which has a positive effect on prolonging the shelf life of fruits. Therefore, the composite film is a commercial application to maintain the weight of fruits in cold storage [64]. Electrostatic spraying chitosan coating can prolong the shelf life of strawberries by 2 days at 4 °C, which is very important for the transportation and sales of strawberries. The SA/AgNPs/APP composite film can prolong the shelf life of strawberries by 8 days, which is not only the excellent barrier performance of SA, but also the super antibacterial activity of AgNPs and the synergistic antibacterial effect and antioxidant capacity of APP [65].

#### 3.9.2. Decay Rate

Figure 8a shows the decay rate of the strawberry samples over a 12-day storage period. During the whole experiment, the decay rate of each group of samples increased with the increase in time. The decay rate of the air control group was the highest in the first 8 days, from 0% to 45.85%. On the 10th day, the decay rate of PE control group was higher than that of the air control group (88.45%). The rotten rate of strawberry fruit covered with the SA/AgNPs/APP composite film was the lowest in the experimental group. On the 12th day, the decay rate (30.45%) was only one third of that of PE. It can be seen that SA/AgNPs/APP composite film can extend the shelf life of strawberry fruit for about 4 days under the condition of 4 °C, and its fresh-keeping effect is significantly better than that of PE film. Kumar et al. [66] added AgNPs into chitosan, gelatin, and polyethylene glycol, the composite film prepared by tape casting can effectively prolong the shelf life of red grapes for 14 days. They proposed that this is related to the strong antibacterial property of nanoparticles and the reduction in WVP value. However, the chitosan film loaded with AgNPs can prolong the shelf life of litchi for 7 days and still maintain a good freshness. This is because AgNPs have a unique nanosize, and the surface effect is easy to penetrate into fungal spores or mycelium, and then destroy the cell wall and its normal metabolism, thus inhibiting the fruit decay caused by microorganisms [43]. Fan et al. used SA coating and *Cryptococcus Lloyd* to keep strawberries fresh at 20 °C. After 5 days of storage, the decay rate of SA/Cryptococcus laurentii/chitosan coating was lower than 25%, which effectively reduced the decay rate of the strawberry fruit by 58%. The main reason is that SA coating forms matrix, sealing the surface of the fruit, and adding antibacterial substances. Clostridium laurentii is the main defense barrier against fungal infection. However, the decay rate of the SA/AgNPs/APP composite film was 30.45% on the 12th day. The reason was that SA had the sealing effect and the antibacterial effect of APP and AgNPs. Low temperature was also an important condition to ensure the low decay rate of the strawberry fruit [63].

#### 3.9.3. Weight Loss

Figure 8b is the trend chart of the weight loss rate during the strawberry test. It can be seen from the chart that the samples of the experimental group and all control groups are increasing, which can be attributed to the loss of mass caused by dry matter consumption and transpiration during fruit metabolism. The water loss caused by transpiration is an important reason for the degradation of horticultural crops, which not only causes the direct loss of quality, but also leads to the decline of appearance, texture, and nutritional quality [67]. Based on this, the air control has the highest weight loss rate, reaching the 27.36%. It should be noted that the PE control had the lowest weight loss rate, which increased slightly from 0% to 5.9% in 12 days. However, the higher decay rate and lower sensory evaluation of PE-coated strawberries should not be ignored. The reason for this trend is that PE has a high barrier, which can block oxygen, carbon dioxide, and water vapor, and inhibit the internal consumption and transpiration of strawberry fruit. The subsequent strawberry decay and deterioration, strawberry internal consumption and transpiration mechanism, as well as weight loss rate maintained a high value [61]. Compared with the SA/APP (13.16%) and SA (15.53%) control group, strawberry fruit coated with the SA/AgNPs/APP composite film had the lowest weight loss rate, and the highest weight loss rate (9.62%) was on the 12th day. First, SA has a certain barrier property. Second, after the addition of AgNPs, the water permeability of the composite film decreased and the barrier property increased, which inhibited the respiration and transpiration of strawberry, resulting in a low weight loss rate. This conclusion is the same as that of Jamróz et al. [68], and AgNPs can also prevent bacterial contamination and delay the weight loss of strawberry fruit. Fan et al.’s results indicate that compared with antimicrobial agents, SA is more effective in providing physical barriers to prevent water loss, thus delaying dehydration and fruit wrinkle. Compared with SA/AgNPs/APP composite film (9.62% weight loss rate on the 12th day), the weight loss rate of SA/Cryptococcus laurentii/chitosan coating was higher, which was 45% on the 5th day. It is concluded that low temperature is an effective means to inhibit respiration and transpiration of fruits and vegetables [63].

#### 3.9.4. Firmness

As shown in Figure 8c, the hardness of the strawberry fruit decreased rapidly to 3.34 N on the 2nd day. In addition, the hardness of the strawberries will also decrease due to decay and softening. The hardness of the SA/APP control group, SA control group, PE control group, and air control group decreased significantly on the 6th, 6th, 6th, and 2nd day, respectively, and finally reached the lowest value on the 12th day at 0.85 N, 0.78 N, 0.6 N, 1.11 N. Interestingly, compared with other control groups, the strawberry in the air control group rotted the most, but the hardness at the end of the test was not the lowest. This is because the strawberry is exposed to 4 C of refrigerator air, the moisture evaporation causes the skin to shrink, and the hardness value obtained by puncturing the strawberry skin is larger due to the dried surface of the strawberry [69]. The hardness of strawberries wrapped in SA/AgNPs/APP was the slowest decreasing trend, and its hardness was always higher than that of all the control groups. On the 10th day, the downward trend becomes faster, and on the 12th day, it reached 1.45 N. Niu et al. [70] suggested that the decrease in hardness was not only related to the loss of water, but also to the degradation of pectin, cellulose, and other cell wall polysaccharides in fruits. The tissue structure and solubilization of pectin and hemicellulose which can be extracted with water are the most important factors to determine the endoplasmic soil of the fruit, and the hardness can reflect the freshness of the fruit [71]. Therefore, SA/AgNPs/APP composite film can effectively maintain the freshness of strawberry fruit, and it has a certain research significance to apply it to strawberry fresh-keeping research. Previous studies have reported that SA/tea polyphenols coating can maintain the hardness of jujube. This may be related to the effect of edible coating on the internal gas composition of jujube, thus reducing the respiration rate and enzyme activity, and finally hindering the reduction in hardness. In the SA/AgNPs/APP study, SA and temperature are important methods to inhibit strawberry respiration and maintain its hardness [72].

#### 3.9.5. Titratable Acidity

Figure 8d shows the changes in the titratable acid content of strawberries in the experimental group and different control groups during storage. As can be seen from the figure, the titratable acid content during storage decreases continuously with the increase in storage days, which may be due to the consumption of organic acids during respiration [73]. The titratable acid content of the air control group decreased the most rapidly, reaching the lowest titratable acid content value of 5.58 mg/100 g on the 12th day. However, the strawberry with the SA/AgNPs/APP composite film coating in the experimental group had the slowest decrease in value from the initial 12.2 mg/100 g reduced to 9.15 mg/100 g. This change trend is similar to that of Han et al. [74]. The decrease in titratable acid content during storage indicates fruit senescence. In this study, the composite film can effectively delay the change of titratable acid content, indicating that the composite-film-coated strawberry delayed the senescence of strawberries. This may be due to the barrier properties of the composite film. The respiration of the strawberry changes the internal interaction and the proportion of gas, leading to delayed ripening and the effect of smart modified atmosphere packaging [74]. In addition, compared to other control groups, AgNPs have strong antibacterial properties and APP has strong anti-oxidation properties, which leads to a decrease in microbial activity and delays the loss of titratable acid caused by microorganisms [73].

#### 3.9.6. pH

During the test, the change of organic acid content in the fresh strawberry fruit can be reflected by pH value. Figure 8e shows the pH value of each group of strawberries during storage. The initial pH value of strawberries is 3.53, and the pH value of the experimental group (SA/AgNPs/APP) and all control groups (SA/APP, SA, PE and air control) increases significantly with the increase in storage time. This trend of pH change is similar to the previous study, which shows that the change trend is caused by fruit aging [74]. The pH value of the air control group and PE control group significantly increased from 3.55 to 4.20 and 4.25, respectively, after 12 days, while that of SA control group and SA/APP control group was only 3.95 and 3.9, respectively, after 12 days. The pH value of SA/AgNPs/APP group increased the most slowly, reaching the lowest pH value of 3.75 after 12 days. As reported by Martínez-Erreret et al. [75], the acidity usually decreases during fruit ripening, which is due to the use of organic acids during the breathing process and conversion to sugar. As most fruits mature, the acidity decreases and the sugar content increases, which is a sign that the fruit is fully ripe. Same as the conclusion of titratable acid, SA/AgNPs/APP composite film can effectively reduce the respiration rate of strawberry, reduce the consumption rate of organic acid, and delay the decreasing trend of acidity [27]. When the strawberry rots and mold grows on it, the microorganism grows in the acid substrate, and its growth, reproduction, metabolism and other life activities will reduce the acidity. From this point of view, the strong antibacterial properties of SA/AgNPs/APP composite film are beneficial, inhibiting the metabolic activity of microorganisms and maintaining pH value [75]. This is consistent with the conclusion of Dhital et al. They observed that the pH increased to 4.13 with the increase in storage days. The increase in pH during storage may be related to the effect of fruit respiration rate, which is attributed to the increase in oxygen content. The main reason why SA/AgNPs/APP composite film inhibits the increasing trend of pH is the barrier property of SA [76].

#### 3.9.7. Soluble Solids and Vitamin C

As shown in Figure 8f, during 12 days of storage, the soluble solids showed a trend of increasing first and then decreasing, which is the same as the research trend of Jiang et al. [65]. Soluble solids generally decreased, which was mainly due to the rapid physiological degradation of the strawberry fruit, and its slight increase was related to the decomposition of the strawberry cell wall, the decrease in respiration rate and the increase in dry matter caused by water loss [65]. In the experimental group, the soluble solids of the strawberry fruit wrapped in SA/AgNPs/APP composite film increased from 8.0% to 9.5% on the 4th day and decreased by about 2.5% on the 12th day. Compared with the soluble solids content (5.0%) on the 12th day of the air control and the soluble solids content (5.2%) of the PE control, the experimental group SA/AgNPs/APP can still maintain a higher soluble solids content of 8.0%. The data show that the SA/AgNPs/APP composite film is more conducive to effectively inhibiting the respiration and metabolism of strawberry life activities, thereby delaying the decay of soluble solids [65]. Compared with the commercially available plastic wrap PE, it is more suitable for strawberry preservation and prolonging the shelf life of strawberries. The soluble solids of strawberry coated with SA/limonene composite film increased to 9.5 from the 2nd day to the 5th day, and decreased to 7.5 on the 9th day, which was not only the same as that of the strawberry coated with SA/AgNPs/APP composite film, but also had no significant difference [76,77].

Vitamin C (Vc) is a minor constituent of strawberries, but it is an important nutrient and has a vital role for humans. In the preservation study (Figure 8g), the content of strawberry Vc covered by composite film showed a downward trend, while the SA/AgNPs/APP film coating treatment greatly inhibited the decrease in Vc content. A decrease in Vc content indicates that the antioxidant activity of the fruit is reduced, which occurs during fruit aging. The results showed that the SA/AgNPs/APP coating treatment can slow down the loss of Vc in strawberries and slow down the senescence of strawberries [73]. It can still maintain the Vc content of 72.17 mg/100 g on the 12th day, with only a decrease of 10.90%. In the study of Khodaei et al., the content of Vc in the control group decreased by 40.25%. In this study, the SA/AgNPs/APP film can retain ascorbic acid by reducing oxygen diffusion and respiratory rate (which can delay the deterioration of oxidation reaction of ascorbic acid in fruits). AgNPs and APP can help maintain vitamins by reducing the number of pathogenic microorganisms and reducing water loss or oxygen diffusion [78]. The higher activity of defense enzymes can quickly eliminate superoxide anion and hydrogen peroxide in fruit tissue, which is very beneficial to delay the senescence of jujube. In the study of the preservation of jujube with SA/apple polyphenols composite film, the activities of superoxide dismutase (SOD), peroxidase (POD) and catalase (CAT) of jujube can be maintained at a high level by adding 1 g/L of tea polyphenols into edible SA coating until the end of storage. Similarly, we can see that APP is of great significance for the maintenance of the activity of active enzymes [72].

## 4. Conclusions

An SA/AgNPs/APP antibacterial and anti-oxidation composite film is prepared by casting film. The effects of the ratio of SA and AgNPs and different ultrasonic times on the performance of the composite film were explored. The results showed that the comprehensive performance of the ratio of 7:3 and ultrasonic time for 30 min was relatively superior. Due to the AgNPs, the film appears black and has strong visible light barrier properties. SA/AgNPs/APP films showed the lowest water vapor permeability value of 0.75 × 10^−11^ g/m·s·Pa. The composite film has good strength and softness, of which TS and E were 23.94 MPa and 29.18%, respectively. SEM images showed that the surface of the composite film was smooth and AgNP distribution was uniform. Additionally, it had excellent antibacterial properties against *E. coli* (92.01%) higher than that of *S.*
*aureus* (91.26%). At the same time, the included APP had strong antioxidant properties (98.39%). In view of the strong antibacterial properties of AgNPs, the strong oxidation resistance of APP and the excellent barrier properties of the composite film, the composite film can effectively extend the shelf life of strawberries under 4 °C for 8 days, and each performance index is better than the SA, SA/APP, PE, and air control groups. This provides a reference for the fixation of AgNPs and expansion of the application range of AgNPs.

## Figures and Tables

**Figure 1 polymers-12-02096-f001:**
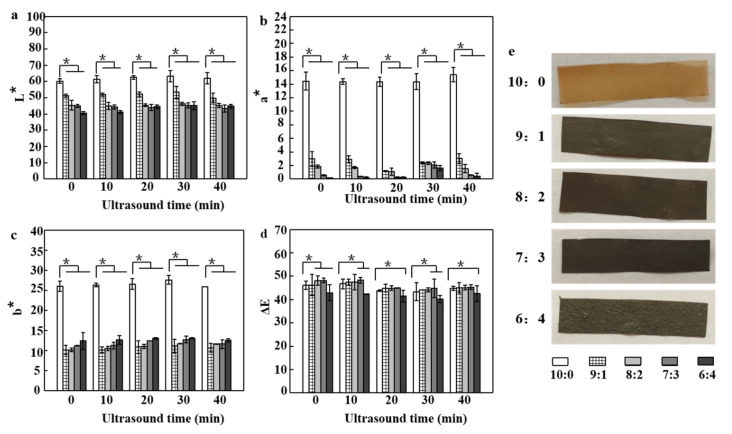
(**a**) L*, (**b**) a*, (**c**) b*, and (**d**) ΔE values of the sodium alginate (SA)/silver nanoparticles (AgNPs)/ apple polyphenol (APP) composite film with different ultrasonic time, respectively; (**e**) the picture of the SA/AgNPs/APP composite films with different proportions. (* *p* < 0.05, *n* = 5.)

**Figure 2 polymers-12-02096-f002:**
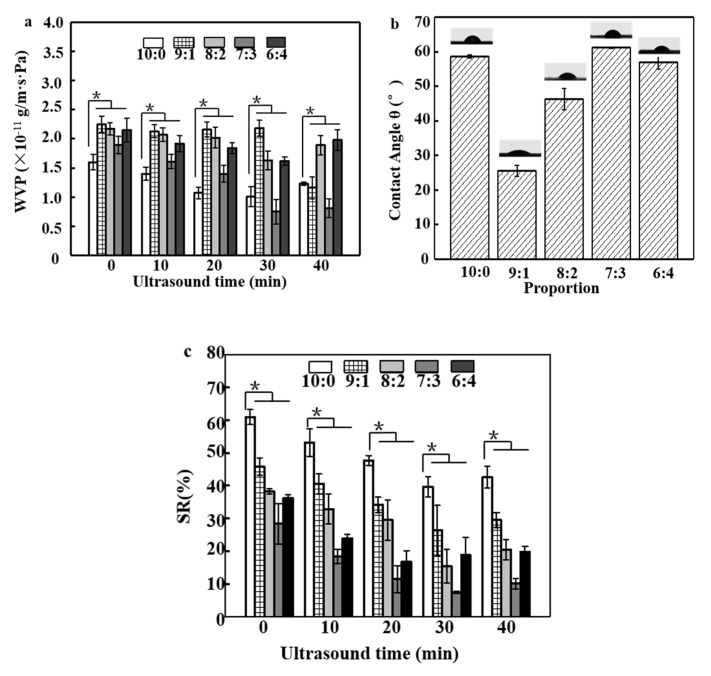
(**a**) Water vapor permeability (WVP) of SA/AgNPs/APP composite film SA/APP and under different ultrasonication times; (**b**) the contact angle of SA/AgNPs/APP (ultrasound time 30 min) composite film with different addition ratios, (**c**) swelling ratio (SR) of film (* *p* < 0.05, *n* = 5).

**Figure 3 polymers-12-02096-f003:**
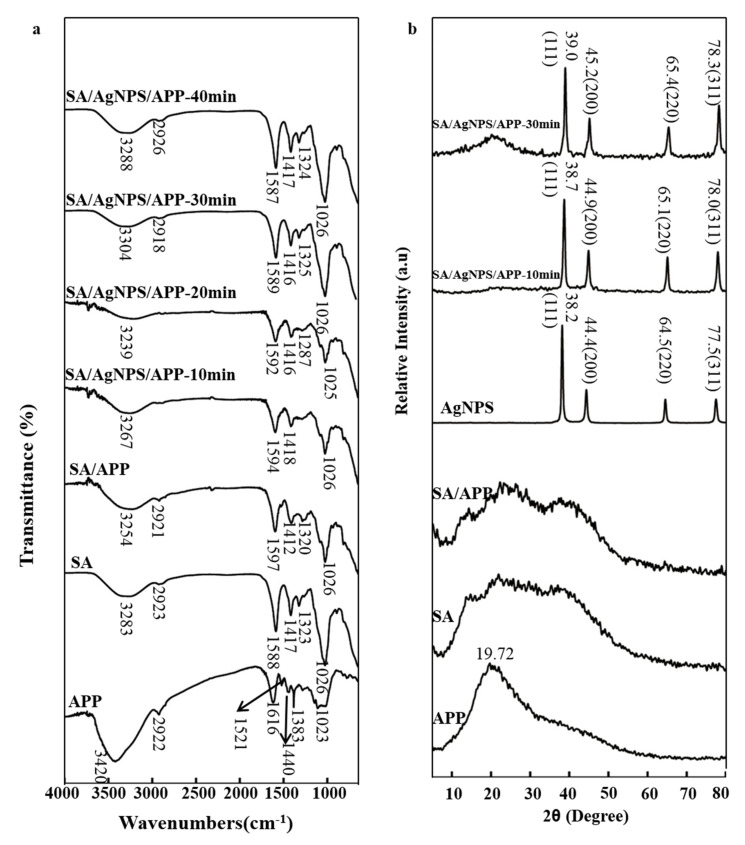
(**a**) FTIR and (**b**) XRD of SA, APP, Ag, SA/APP and SA/AgNPs/APP composite films.

**Figure 4 polymers-12-02096-f004:**
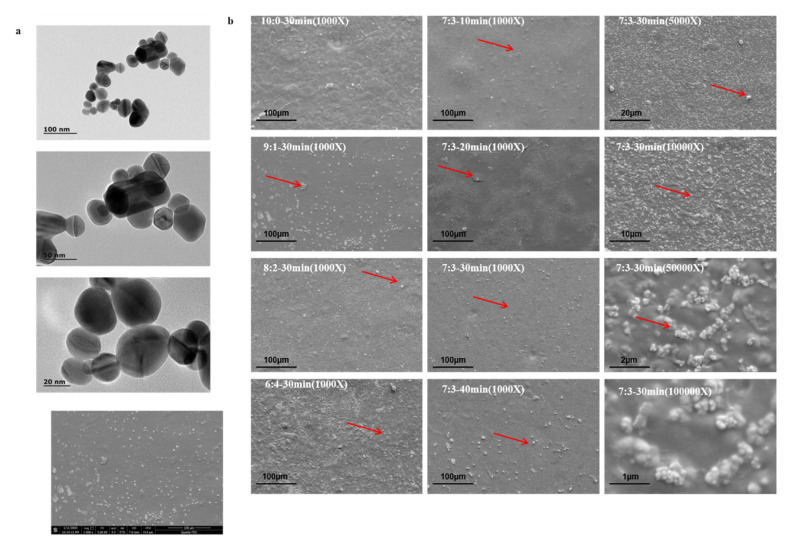
(**a**) TEM of AgNPs; (**b**) SEM images of SA/AgNPs/APP composite film with different ultrasound time, different addition ratio and different magnification.

**Figure 5 polymers-12-02096-f005:**
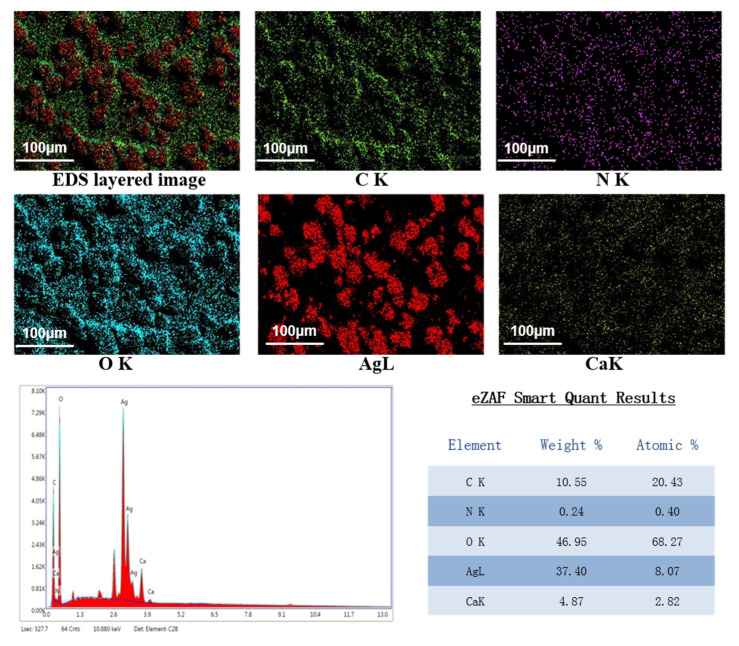
EDS image of SA/AgNPs/APP composite film.

**Figure 6 polymers-12-02096-f006:**
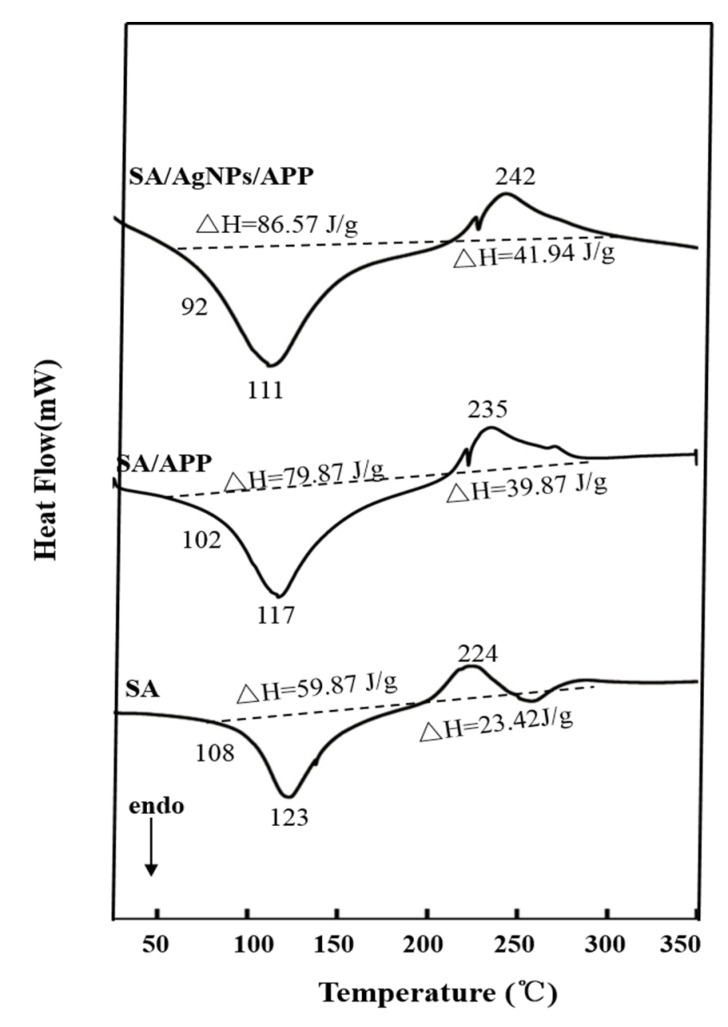
DSC of SA, SA/APP and SA/AgNPs/APP composite films.

**Figure 7 polymers-12-02096-f007:**
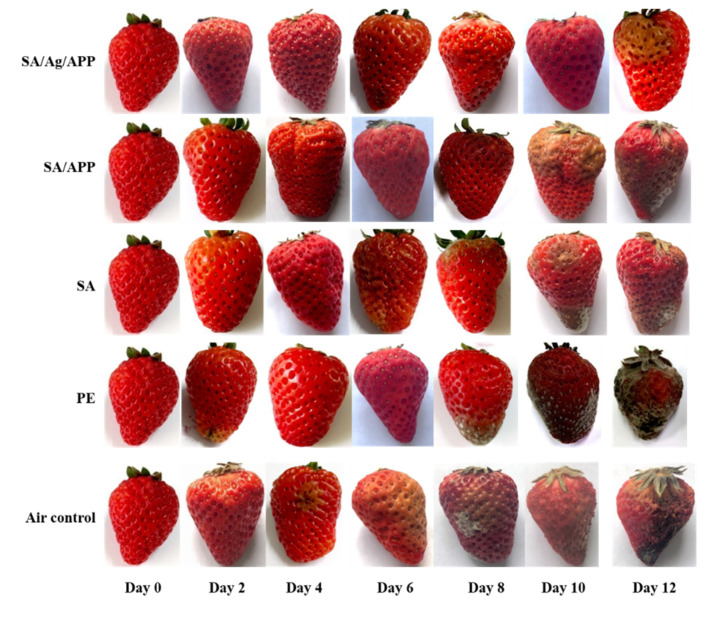
Picture of strawberries during 12 days of strawberry storage.

**Figure 8 polymers-12-02096-f008:**
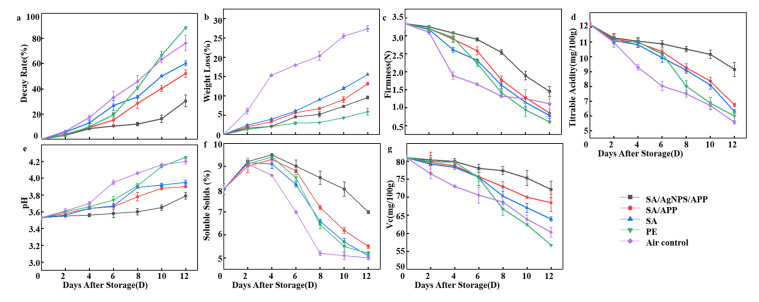
(**a**) Decay rate, (**b**) weight loss, (**c**) firmness, (**d**) titratable acidity, (**e**) pH, (**f**) soluble solids and (**g**) V_C_ of strawberries during 12 storage.

**Table 1 polymers-12-02096-t001:** Decay rate evaluation.

Decay Level	Evaluation Criterion
1	1–20% of surface was infected
2	21–40% of surface was infected
3	41–60% of surface was infected
4	61–80% of surface was infected
5	≥81% of surface was infected

**Table 2 polymers-12-02096-t002:** Mechanical properties of SA/AgNPs/APP film.

	SA:AgNPs	0 min	10 min	20 min	30 min	40 min
TS (MPa)	10:0	23.16 ± 1.64 ^a^	25.72 ± 1.53 ^a^	27.96 ± 1.41 ^a^	28.06 ± 1.55 ^a^	24.07 ± 1.28 ^a^
	9:1	21.46 ± 1.38 ^b^	23.23 ± 1.15 ^b^	24.31 ± 1.21 ^b^	26.98 ± 1.32 ^b^	23.43 ± 1.19 ^a^
	8:2	20.04 ± 1.12 ^b^	22.49 ± 1.13 ^b^	23.96 ± 1.13 ^b^	24.07 ± 1.18 ^b^	21.52 ± 1.10 ^b^
	7:3	18.11 ± 0.99 ^c^	20.54 ± 1.07 ^b^	22.57 ± 1.10 ^b^	23.94 ± 1.13 ^b^	20.80 ± 1.02 ^b^
	6:4	16.34 ± 0.89 ^d^	17.45 ± 0.92 ^c^	20.24 ± 0.96 ^c^	21.50 ± 1.01 ^c^	16.67 ± 0.90 ^c^
E (%)	10:0	18.10 ± 1.12 ^a^	21.60 ± 1.19 ^a^	24.31 ± 1.23 ^a^	26.10 ± 1.39 ^a^	23.30 ± 1.11 ^a^
	9:1	13.99 ± 2.19 ^b^	22.83 ± 1.14 ^a^	25.95 ± 1.26 ^a^	27.49 ± 1.42 ^a^	26.51 ± 1.32 ^b^
	8:2	14.79 ± 1.16 ^b^	23.25 ± 1.18 ^a^	26.83 ± 1.37 ^a^	28.90 ± 1.54 ^a^	27.95 ± 1.40 ^b^
	7:3	16.40 ± 1.18 ^a^	25.15 ± 1.21 ^b^	27.44 ± 1.40 ^a^	29.18 ± 1.59 ^a^	28.29 ± 1.46 ^b^
	6:4	21.08 ± 1.31 ^c^	23.15 ± 1.18 ^a^	25.07 ± 1.34 ^a^	27.09 ± 1.41 ^a^	26.04 ± 1.33 ^b^

Different letters indicate significant differences in each column of data (*p* < 0.05, *n* = 5); 0–40 min is the ultrasonic processing time. TS, tensile strength; E, elongation at break.

**Table 3 polymers-12-02096-t003:** Antimicrobial and antioxidant activities of the composite films.

Samples	Antibacterial Property (%)		DPPH (%)
	*S. aureus*	*E. coli*	
SA	-	-	1.01 ± 0.53
SA/AgNPs/APP-10:0	27.61 ± 0.53 ^d^	27.84 ± 0.49 ^d^	99.89 ± 0.07 ^a^
SA/AgNPs/APP-9:1	58.98 ± 0.28 ^c^	54.90 ± 0.22 ^c^	95.57 ± 0.32 ^a^
SA/AgNPs/APP-8:2	83.17 ± 0.73 ^b^	84.54 ± 0.53 ^b^	97.23 ± 0.21 ^a^
SA/AgNPs/APP-7:3	91.26 ± 0.49 ^a^	92.01 ± 0.55 ^a^	98.39 ± 0.11 ^a^
SA/AgNPs/APP-6:4	92.48 ± 0.88 ^a^	92.78 ± 0.37 ^a^	95.92 ± 0.35 ^a^

Different lowercase letters indicate significant differences in each column of data (*p* < 0.05, *n* = 5).

**Table 4 polymers-12-02096-t004:** Sensory evaluation of strawberry fruit.

Days (d)	0	2	4	6	8	10	12
SA/AgNPs/APP	9.0 ± 0.1 ^a^	8.5 ± 0.3 ^a^	8.2 ± 0.1 ^a^	7.9 ± 0.1 ^a^	7.3 ± 0.2 ^a^	7.0 ± 0.5 ^a^	6.5 ± 0.2 ^a^
SA/APP	9.0 ± 0.1 ^a^	8.2 ± 0.2 ^a^	7.8 ± 0.4 ^b^	7.3 ± 0.3 ^a^	5.5 ± 0.1 ^b^	4.5 ± 0.6 ^b^	2.5 ± 0.5 ^b^
SA	9.0 ± 0.1 ^a^	8.0 ± 0.1 ^a^	7.6 ± 0.7 ^b^	6.8 ± 0.2 ^b^	4.2 ± 0.2 ^c^	2.5 ± 0.4 ^c^	1.0 ± 0.2 ^c^
PE	9.0 ± 0.1 ^a^	8.4 ± 0.3 ^a^	7.5 ± 0.2 ^b^	6.3 ± 0.4 ^b^	4.0 ± 0.4 ^c^	1.0 ± 0.2 ^d^	1.0 ± 0.3 ^c^
Air control	9.0 ± 0.1 ^a^	7.8 ± 0.1 ^b^	6.0 ± 0.1 ^c^	4.2 ± 0.3 ^c^	3.1 ± 0.2 ^d^	1.3 ± 0.3 ^d^	1.1 ± 0.1 ^c^

Different lowercase letters indicate significant differences in each column of data (*p* < 0.05, *n* = 5).

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
