# Peer review of "Investigation of Ultrasonic Treatment on Physicochemical, Structural and Morphological Properties of Sodium Alginate/AgNPs/Apple Polyphenol Films and Its Preservation Effect on Strawberry"

_polymers, 2020, doi:10.3390/polym12092096_

Round 1

Reviewer 1 Report

The manuscript aimed to investigate the ultrasonic treatment on properties of sodium alginate/AgNPs/apple polyphenol films and its preservation effect on strawberry.

The paper is generally well written and discussed, but some misunderstanding issues must be clarified:

  • Page 2, lines 55 and 73: remove the space;
  • Page 3, line 122: remove the dot after [17];
  • Page 5, table 1: “1%-20%” instead of “1%-2%”;
  • Page 8, line 298: remove the dot after [31];
  • Page 9, line 354: “1588” instead of “1588-1”;
  • Page 13, line 453: “APP” instead of “app”;
  • Page 14, line 463: “ coli (27.84± 463 0.49%) and S. aureus (27.61±0.53%)” instead of “E. coli (27.61±0.53%) and S. aureus (27.84± 463 0.49%)”;
  • Page 15, table 3: “SA/AgNPs/APP-10:0” instead of “SA/AgNPs/APP-10:0”;
  • Page 15, line 513: “6 days after storage” instead of “6 days before storage”;
  • Page 18, line 623: “Soluble solids and vitamin C” instead of “Soluble solids and Vc”;
  • Page 18, line 637: “Vitamin C (Vc)” instead of “Vc”;
  • Uniformize the letter of significance level along with the text, which is, “p” or “P”;
  • The meaning of TS (MPa) and E(%) present in table 2 should be specified elsewhere.

After these issues have been tackled, I recommend the paper for publication in Polymers.

Author Response

Comments and Suggestions for Authors 1:

  1. The manuscript aimed to investigate the ultrasonic treatment on properties of sodium alginate/AgNPs/apple polyphenol films and its preservation effect on strawberry. The paper is generally well written and discussed, but some misunderstanding issues must be clarified:

Re: Thanks.

  1. Page 2, lines 55 and 73: remove the space;

Re: Done, thank you.

  1. Page 3, line 122: remove the dot after [17];

Re: Deleted. Thanks.

  1. Page 5, table 1: “1%-20%” instead of “1%-2%”;

Re: ‘1%-2%’ has been changed to ‘1%-20%’, thank you.

  1. Page 8, line 298: remove the dot after [31];

Re: Done, thanks.

  1. Page 9, line 354: “1588” instead of “1588-1”;

Re: Unit (cm-1) of 1588 has been added. Thanks.

  1. Page 13, line 453: “APP” instead of “app”;

Re: It has been changed to upper case. Thanks

  1. Page 14, line 463: “coli (27.84± 463 0.49%) and S. aureus (27.61±0.53%)” instead of “E. coli (27.61±0.53%) and S. aureus (27.84± 463 0.49%)”;

Re: The data has been corrected. Thank you.

  1. Page 15, table 3: “SA/AgNPs/APP-10:0” instead of “SA/AgNPs/APP-10:0”;

Re: Corrected, thanks.

  1. Page 15, line 513: “6 days after storage” instead of “6 days before storage”;

Re: The mistake has been corrected. Thank you.

  1. Page 18, line 623: “Soluble solids and vitamin C” instead of “Soluble solids and Vc”;

Re: The complete form of Vc has been completed, thanks.

  1. Page 18, line 637: “Vitamin C (Vc)” instead of “Vc”;

Re: Corrected, thanks.

  1. Uniformize the letter of significance level along with the text, which is, “p” or “P”;

Re: The letter of significance level in the full text has been unified into ‘P’.

  1. The meaning of TS (MPa) and E(%) present in table 2 should be specified elsewhere.

After these issues have been tackled, I recommend the paper for publication in Polymers.

Re: TS and E are the tensile strength and elongation at break of the film respectively. I have added a note to line 122 in the article. If you have any other questions, please point them out. I will revise them carefully. Thanks.

Reviewer 2 Report

  1. The abstract can be improved: it should should contain film composition(all components), a list of the methods used, which led to the results presented. I also think that a short paragraph referring to the process by which it was concluded that the film has beneficial effects on strawberries should be integrated.
  2. The introductory part is too short. For such an abundance of methods and determinations made, it is necessary to support the statements with scientific support. Also, to carry out this process, more bibliographic references are needed than those used by the authors.
  3. Lines 58-60, The antibacterial activity of aureus and E. coli showed that the fiber had strong antibacterial activity” ?????,   this is an error. 
  4. Lines 60-61, It compares the films have different base compositions.
  5. Line 88, Glycerol(used as a good plasticizer) is mentioned in the Materials and Methods chapter, but it is not mentioned in the composition of the film in the abstract.
  6. Lines 92-109, It is described the film obtaining, without the glycerol plasticizer, mentioned in the Materials chapter. What was the plasticizer used?
  7. Lines 134-135 - the determination made actually refers to the moisture content. It is measured and determined according to the information noted in the article. Swelling capacity or swelling ratio is calculated according to the formula :     SR, (%)= [(Wt-W0)/W0] x 100

where SR represent the swelling ratio,  Wt  is the film mass at t moment, and W0  represent the initial mass (g). Samples are weighed before and after immersion and they are not subjected to drying but only the excess water is removed by lightly tapping with filter paper. The moisture content determination implies the steps presented into the manuscript and is calculated according with formula presented. You can see and observe all these differences in paper „Rethinking the Future of Food Packaging: Biobased Edible Films for Powdered Food and Drinks”, https://doi.org/10.3390/molecules24173136.

  1. Lines 166-174: it is not specified how the strawberries were covered with the obtained film: the fructs were directly coated or were kept in trays made of another material and then covered with film? Please specify this. If the fruits has been covered, specify how to carry out the coating and drying process.
  2. Comparisons are made with other results obtained in other specialized works, but they do not refer to the same methods or treatments used, to the same types of film or film-coated products (for example, lines 391-398, lines 538-547).
  3. Line 460, figure 6, Why using a temperature range from 50 to 350°C in thermal analysis if you are interested only in the temperature range in which they are stored the strawberries?.
  4. Lines 498 – 528: the results presented can be supported by images of the products. Also, the results of the sensory evaluation could have been highlighted differently: for example, through graphs, easy to follow and which would have presented more clearly the differences obtained by the tested products.
  5. Line 548/567: the results should be stated with fruits images.
  6. The weakest parts of the manuscript are Conclusions and Abstract. Thus, it needs improvement.
  7. The manuscript can be supported with several data from specialized works. I consider that 32 bibliographic references are few for a work of such complexity.
  8. The Englih level should be improved. Throughout the work there are prepositions and connecting words used incorrectly, which prevents the correct understanding of the text and often changes its meaning.

Author Response

Comments and Suggestions for Authors 2:

  1. The abstract can be improved: it should should contain film composition(all components), a list of the methods used, which led to the results presented. I also think that a short paragraph referring to the process by which it was concluded that the film has beneficial effects on strawberries should be integrated.

Re: The abstract part has been rewritten as follows. Thank you. An antibacterial and anti-oxidation composite film was prepared by casting method using sodium alginate (SA) and apple polyphenols (APP) as the base material and glycerol as plasticizer. Silver nanoparticles (AgNPs) were deposited by ultrasonic-assisted electrospray method. The degree of influence of the addition ratio of SA and AgNPs and different ultrasonic time on the mechanical properties, barrier properties, optical properties, and hydrophilicity of the composite film was explored. The composite fims were characterized by Fourier transform infrared spectroscopy (FTIR), X-ray diffraction (XRD) and scanning electron microscopy (SEM). The results showed that SA: AgNPs ratio of 7:3 and the ultrasonic time for 30 minutes has the best comprehensive performance, SA/AgNPs/APP films showed the lowest water vapor permeability value of 0.75×10-11 g/m·s·Pa. The composite film has good strength and softness, which tensile strength (TS) and elongation at break (E) were 23.94 MPa and 29.18%, respectively. SEM images showed that the surface of the composite film was smooth and AgNPs distribution was uniform. The composite membrane showed broad antibacterial activity, and the antibacterial activity of E. coli (92.01%) was higher than that of S. aureus (91.26%). However, due to the addition of APP, its antioxidant activity can reach 98.39%, which has synergistic effect on antibacterial activity. Strawberry as a model, the results showed that this composite film can prolong the shelf life of strawberry for about 8 days at 4°C, effectively maintain the storage quality of strawberry. Compared with the commonly used PE film on the market, it has more fresh-keeping effect and can be used as an active food packaging material. Thanks.

  1. The introductory part is too short. For such an abundance of methods and determinations made, it is necessary to support the statements with scientific support. Also, to carry out this process, more bibliographic references are needed than those used by the authors.

Re: I have expanded the introduction, adding more descriptions of sodium alginate, apple polyphenols and nano silver, which made the logic and content of the article more fluent and full. Thank you.

  1. Lines 58-60, „The antibacterial activity of aureus and E. coli showed that the fiber had strong antibacterial activity” ?????, this is an error.

Re: We have corrected the error of the expression, as follows, thank you. The results of antibacterial experiments against S. aureus and E. coli showed that the fiber had strong antibacterial activity.

  1. Lines 60-61, It compares the films have different base compositions.

Re: In this study, AgNPs were captured by polyvinyl alcohol (PVA) and sodium alginate (PVA). The loaded silver nanoparticles showed high bactericidal activity and thermal stability, and encapsulated AgNPs well. It also indirectly reflected the ability of sodium alginate and polyvinyl alcohol to capture nanoparticles, indicating that sodium alginate has the ability to capture nanoparticles. On the basis of previous studies, the encapsulation ability of sodium alginate was comprehensively explained, but all the studies could not be completed by single sodium alginate, and the synergistic capture had an effective effect (Food Chem., 2017, 234,103-110.). I have revised the wording of this part. Thanks.

  1. Line 88, Glycerol(used as a good plasticizer) is mentioned in the Materials and Methods chapter, but it is not mentioned in the composition of the film in the abstract.

Re: This part has been added to the abstract. You can also see this change in the last paragraph of the introduction. Thank you.

  1. Lines 92-109, It is described the film obtaining, without the glycerol plasticizer, mentioned in the Materials chapter. What was the plasticizer used?

Re: We used glycerin as plasticizer. We modified the film preparation process and added this part. Thank you. The SA/AgNPs/APP composite film was prepared by a casting method. The solution was made of APP (1%), glycerol (2%) and different mass ratios of SA:AgNPs (total mass 2%), i.e., 10:0, 9:1, 8:2, 7:3, 6:4. First, different amount of SA was dissolved in distilled water to get SA aqueous solution. APP powder and glycerol were added and mixed using a booster stirrer (JJ-1, Changzhou Jintan Sanhe Instrument Co., Ltd.) to fully dissolve and mix, thus obtaining a precursor solution. The solution was cast on a glass plate and dried in a digital display air drying oven (RZ101-1, Suzhou Runze Oven Manufacturing Co., Ltd.) at 50°C for 6 h. After obtained SA/APP films, the films were fixed as the collector for electrospray. AgNPs powders were added to 5 mL of absolute ethanol and ultrasonically dispersed using a XH-2008D ultrasonic instrument (Xianghu development Co., Ltd., Beijing, China) equipped with a reactor with a thermostatic water bath (temperature accuracy of ±1 °C), a mechanical stirrer and a microtip probe (diameter of 8 mm) for 0, 10, 20, 30, and 40 min (frequency: 40 kHz, power: 50 W). Subsequently, the suspension was filtered through a 5 mL syringe with a single needle configuration (200 μm outside diameter and 100 μm inside diameter) and was continuously pushed by a syringe pump (Zhejiang University Medical Instrument, Hangzhou, China) at 0.20 mL/h, using a electrospray apparatus equipped with 10 kV power supply (Tianjing High-Voltage Power Supply Co., Tianjing, China), as reported previously [15]. Spraying distances of 5 cm were set between the syringe nozzle and the SA/APP films. Finally, the composite film was soaked to a 2% calcium chloride aqueous solution for cross-linking (2 minutes), and then air-dried to obtain a SA/AgNPs/APP composite film. Thanks.

  1. Lines 134-135 - the determination made actually refers to the moisture content. It is measured and determined according to the information noted in the article. Swelling capacity or swelling ratio is calculated according to the formula : SR, (%)= [(Wt-W0)/W0] x 100 where SR represent the swelling ratio,  Wt  is the film mass at t moment, and W0  represent the initial mass (g). Samples are weighed before and after immersion and they are not subjected to drying but only the excess water is removed by lightly tapping with filter paper. The moisture content determination implies the steps presented into the manuscript and is calculated according with formula presented. You can see and observe all these differences in paper „Rethinking the Future of Food Packaging: Biobased Edible Films for Powdered Food and Drinks”, https://doi.org/10.3390/molecules24173136.

Re: Thank you very much for your reminding. I have carefully understood and calculated the index. We confirm that the test item is swelling ratio. We have modified the method test and corresponding analysis part, as follows. Thank you.   

The dried film sample (5×5 cm) was immersed in 50 mL distilled water and kept at room temperature for 24 h, and then samples are weighed before and after immersion (Molecules, 2019, 24, 3136.). The swelling ratio (SR) of the film was calculated using the following formula:     (2)

where Wt is the film mass at t moment (g), and W0 represent the initial mass (g).

The effect of the ratio of SA with AgNPs and ultrasonic treatment time in the swelling-ratio  behavior properties of SA/AgNPs/APP films was evaluated, and the results are shown in Figure 2c. SR of SA/AgNPs/APP films were 149.82% (10:0), 139.82% (9:1), 90.29% (8:2) and 86.23% (6:4), respectively, showing the high water hydrophilic characteristic. After added AgNPs, the minimum SR of the SA/AgNPs/APP films was reached to 13.37% (decreased by 136.45%) in 7:3. The high values of SR of the SA/APP (10:0) films may be attributed to the strong hydrophilicity of SA and APP that can easily interact with water molecules (Carbohyd. Polym., 2017, 163, 81-91.). The decrease in SR indicated water hydrophobic of the SA/APP films were enhanced for the hydrophobic characterization of AgNPs. These differences in the SR changes could be due to the SA/APP content used in each formulation. First, increasing the SA/APP concentration results in a higher number of reactive sites to interact with water, thus increasing the water absorption. On the contrary, it shows why the concentration of AgNPs increases while the SR value decreases. The number of -OH in SA/APP film decreased after incorporation of AgNPs, thus it could combine less water (Food Packaging Shelf., 2018, 18, 157-163.). In addition, this may be due to the interaction between SA polymer chain and AgNPs, which limits the movement chain of SA polymer (Food Chem., 2019, 283, 397-403.). It is worth noting that ultrasonic treatment can significantly reduce the SR of the composite film. When it is 7:3-30 min, the lowest SR is 4.52%. Due to the special behavior of the grain boundary, the hydrophobic repulsion of AgNPs is caused, while the ultrasonic wave disperses AgNPs uniformly in the matrix for a certain time, which hinders the swelling caused by water molecules. Secondly, the uniform AgNPs has certain resistance to the invasion of water molecules, which eventually leads to the decrease of SR. However, the ultrasonic time transition leads to the aggregation of AgNPs and the formation of pores, which results in the increase of SR (Ultrason. Sonochem., 2019, 59, 104731.). As for the decrease of 10:0 SR caused by ultrasound, it may be due to the more closed matrix caused by ultrasound, which makes the blend film more difficult to contact water, and the surface crosslinking degree of CaCl2 is high (Food Packaging Shelf., 2018, 18, 157-163.). Thanks.

  1. Lines 166-174: it is not specified how the strawberries were covered with the obtained film: the fructs were directly coated or were kept in trays made of another material and then covered with film? Please specify this. If the fruits has been covered, specify how to carry out the coating and drying process.

Re: Strawberries were directly coated with the film of the experimental film and the control film (LWT, 113, 108297.). I added this description in the method test section. Thanks.

  1. Comparisons are made with other results obtained in other specialized works, but they do not refer to the same methods or treatments used, to the same types of film or film-coated products (for example, lines 391-398, lines 538-547).

Re: I've made some changes. But I have to point out that there are a lot of studies on strawberry preservation, but there are very few studies on the application of nano silver composite film in strawberry preservation. If there are no similar functional substances between the two, the significance of the comparison is limited.     Secondly, we can see that many studies only compare with their own research objects, because the comparison needs to ensure that the irrelevant conditions are the same. However, in my own research, this condition is satisfied, and the comparison with others' research will be affected by multiple independent variables. But I will try my best to modify and add this part of the analysis. You can see it in line 574-600. Thanks.

  1. Line 460, figure 6, Why using a temperature range from 50 to 350°C in thermal analysis if you are interested only in the temperature range in which they are stored the strawberries?.

Re: First of all, the purpose of testing DSC must be clearly stated. The changes of molecular structure, aggregation structure and molecular motion of materials are studied by detecting the changes of thermophysical properties of samples with temperature or time. DSC is to study the change of heat resistance of the film due to the addition of various materials. This is not in conflict with the application of low temperature preservation of strawberry, and the preservation research is a further study of film.    

In addition, as a model material, strawberry preservation research does not mean that the film can only be applied to strawberry. If its stability at 4 is satisfied, its application range is very narrow.                                             

Secondly, assuming that the film can only be applied to strawberry, its physical and chemical properties must be determined before it is applied to real life. Thermal stability is a very important index to determine whether it is stable. The thermal stability is not only the stability of a certain temperature, but also a temperature range, which should also include the stability of thin films at extremely high temperatures.

Moreover, the main component of the film is sodium alginate. Many studies have reported that the weight loss of pure sodium alginate polymer is mainly divided into two stages. The first stage is the weight loss of pure sodium alginate polymer at 200°C, which is called the water evaporation stage. The second stage is weight loss between 250°C and 500°C due to depolymerization of the polysaccharide network and complex dehydration (Polym. Test., 2020, 81, 106183.). However, due to the limitation of our instruments, the temperature can only reach 400, but it does not affect the thermal stability of the films (Carbohydr. Polym., 2017, 170, 264-270.). Thanks. 

  1. Lines 498 – 528: the results presented can be supported by images of the products. Also, the results of the sensory evaluation could have been highlighted differently: for example, through graphs, easy to follow and which would have presented more clearly the differences obtained by the tested products.

Re: I added images of strawberry in Figure 7 and expressed the sensory evaluation in Table 4. Thanks.

  1. Line 548/567: the results should be stated with fruits images.

Re: Added, thanks.

  1. The weakest parts of the manuscript are Conclusions and Abstract. Thus, it needs improvement.

Re: I have rewritten the Conclusions and Abstract, as follows, thank you.

Abstract: An antibacterial and anti-oxidation composite film was prepared by casting method using sodium alginate (SA) and apple polyphenols (APP) as the base material and glycerol as plasticizer. Silver nanoparticles (AgNPs) were deposited by ultrasonic-assisted electrospray method. The degree of influence of the addition ratio of SA and AgNPs and different ultrasonic time on the mechanical properties, barrier properties, optical properties, and hydrophilicity of the composite film was explored. The composite fims were characterized by Fourier transform infrared spectroscopy (FTIR), X-ray diffraction (XRD) and scanning electron microscopy (SEM). The results showed that SA: AgNPs ratio of 7:3 and the ultrasonic time for 30 minutes has the best comprehensive performance, SA/AgNPs/APP films showed the lowest water vapor permeability value of 0.75×10-11 g/m·s·Pa. The composite film has good strength and softness, which tensile strength (TS) and elongation at break (E) were 23.94 MPa and 29.18%, respectively. SEM images showed that the surface of the composite film was smooth and AgNPs distribution was uniform. The composite film showed broad antibacterial activity, and the antibacterial activity of E. coli (92.01%) was higher than that of S. aureus (91.26%). However, due to the addition of APP, its antioxidant activity can reach 98.39%, which has synergistic effect on antibacterial activity. Strawberry as a model, the results showed that this composite film can prolong the shelf life of strawberry for about 8 days at 4°C, effectively maintain the storage quality of strawberry. Compared with the commonly used PE film on the market, it has more fresh-keeping effect and can be used as an active food packaging material.

Conclusions: An SA/AgNPs/APP antibacterial and anti-oxidation composite film is prepared by casting film. The effects of the ratio of SA and AgNPs and different ultrasonic times on the performance of the composite film were explored. The results showed that the comprehensive performance of the ratio of 7: 3 and ultrasonic time for 30 min was relatively superior. Due to AgNPs, the film appears black and has strong visible light barrier properties. SA/AgNPs/APP films showed the lowest water vapor permeability value of 0.75×10-11 g/m·s·Pa. The composite film has good strength and softness, which TS and E were 23.94 MPa and 29.18%, respectively. SEM images showed that the surface of the composite film was smooth and AgNPs distribution was uniform. And it had excellent antibacterial properties against E. coli (92.01%) was higher than that of S. aureus (91.26%). At the same time, the included APP had strong antioxidant properties (98.39%). In view of the strong antibacterial properties of AgNPs, the strong oxidation resistance of APP and the excellent barrier properties of the composite film, the composite film can effectively extend the shelf life of strawberries under 4°C for 8 days, and each performance index is better than the SA, SA/APP, PE, and air control groups. This provides a reference for the fixation of AgNPs and expansion of the application range of AgNPs. Thanks.

  1. The manuscript can be supported with several data from specialized works. I consider that 32 bibliographic references are few for a work of such complexity.

Re: I deleted a lot of literature in the process of writing, in view of this, I will complete it, thank you.

  1. The Englih level should be improved. Throughout the work there are prepositions and connecting words used incorrectly, which prevents the correct understanding of the text and often changes its meaning.

Re: Sorry, we have revised the full text of the article again by high-level English personnel according to your advice. If you find any problems, please point out that we will continue to revise. Thank you.

Reviewer 3 Report

Please, see attached.

Author Response

Comments and Suggestions for Authors 3: 

Manuscript polymers-906844 entitled “Investigation of ultrasonic treatment on physicochemical, structural and morphological properties of sodium alginate/AgNPs/apple polyphenol films and its preservation effect on strawberry”
written by: Wenting Lan, Siying Li, Yuqing Zhao, Dur E Sameen, Li He and Yaowen
Liu provide inventive idea, applicable product - composite film based on natural polymer sodium alginate (SA), silver nanoparticles (AgNP) and apple polyphenols (APP). As a result of a comprehensive characterization and analysis, authors singled out a sample of composite with the most desirable performances. This SA/AgNP/APP composite film sample can be used as an active food packaging material for prolongation the shelf life of strawberry for about 8 days at 4°C.

This manuscript can be published in the journal "Polymers" after following improvements: 

Re: Thank you very much for your affirmation. I will try my best to modify it. Thank you.

  1. It will be needed to include analysis and comparison of obtained results with similar papers:

Luo, Y., Liu, H., Yang, S., Zeng, J., & Wu, Z. (2019). Sodium Alginate-Based Green Packaging Films Functionalized by Guava Leaf Extracts and Their Bioactivities. Materials, 12(18), 2923.

Sharma, S., Sanpui, P., Chattopadhyay, A., & Ghosh, S. S. (2012). Fabrication of antibacterial silver nanoparticle—sodium alginate–chitosan composite films. Rsc Advances, 2(13), 5837-5843.

Re: Has been added, you can see in lines 336, 468, 481, 340, 387, 465, 583 and 604, thank you.

  1. Please, avoid 1st person plural and rewrite sentence in the page 4 to the 3rd person plural (Liness 169, 472).

Re: Done, thanks.

  1. It would be better to divide Table 2. in two parts: first part for Tensile Strength, and second part for Elongation at break below TS.

Re: Revised, thank you.

  1. Please, it is needed to provide full names of acronyms on first appearance in the main manuscript text and figure caption (e.g. for TS and E, as well as for a, b, c and d in superscripts, WS).

Re: Corrected, thanks.

  1. Please provide analysis about which of the samples is providing better thermal stability in the phase transition for exothermic peaks at about 224-242oC. (lines 457-459)

Re: Added, thank. The exothermic peaks of SA, SA/APP and SA/AgNPs/APP composite films are at 224°C, 235°C, and 242°C, respectively (Polym. Test., 2020, 81, 106183.). Over 200, the polysaccharides began to decompose, including random cleavage of glycosidic bonds, evaporation and elimination of volatile products (Themochmica Acta 1991, 176, 63-68.). APP showed higher carbohydrate content and higher molecular weight, which may be related to the more compact and complex macromolecular structure and more stable to thermal degradation. The research of Donato et al. also showed this result (Carbohydrate Polymers 2020, 229, 115427.). On the other hand, due to the strong hydrogen bonding between alginate and AgNPs, the introduction of AgNPs slightly increased the residue and enhanced the thermal degradation temperature (Desalination, 2020, 485, 114465.).

  1. Is it possible to provide the enthalpy changes data (∆H) of presented phase transitions?

Re: Added, thanks.

  1. Please, check/confirm if the ordinate name label is VC or Vc in Figure 7e? (line 531).

Re: Revised, thank you.

Round 2

Reviewer 2 Report

  1. Lines 58-64, A comparison with chitosan is not necessary, as it is a good antimicrobial agent.
  2. Lines 60-61, It compares the films have different base compositions.
  3. Lines 134-135 – Modifying the formula for Swelling capacity or swelling ratio is not enough, the graphical representations also had to be modified.
  4. Lines 222-223, it is not specified how the strawberries were covered with the obtained film: „Strawberries were directly coated with the film of the experimental film and the control film”??, the fruits were directly coated or were kept in trays made of another material and then covered with film? Please specify this. If the fruits has been covered, specify how to carry out the coating and drying process. If they were packaged in trays, then the antioxidant activity of APP is not relevant.
  5. Line 460, figure 6, Why using a temperature range from 50 to 350°C in thermal analysis if you are interested only in the temperature range in which they are stored the strawberries?.
  6. Lines 583-586, Discussions about polyphenols extracted from guava leaves that have a different composition and cannot be compared with apple polyphenols,  " and that polyphenol mixtures are usually much more active than any single  polyphenol alone [37]”. ??
  7. Lines 614-618, they are not necessary and do not answer the subject.
  8. Lines 691-698, Comparisons are made with other results obtained in other specialized works, but they do not refer to the same methods or treatments used, to the same types of film or film-coated products or the same fruits.

Author Response

Comments and Suggestions for Authors 2:

1. Lines 58-64, A comparison with chitosan is not necessary, as it is a good antimicrobial agent.

2. Lines 60-61, It compares the films have different base compositions.

Re: I'm really sorry. You're right. There's no need to compare it with chitosan film, but APP is effective in enhancing the physical properties, antibacterial and antioxidant properties of the film, so we modified this sentence. Thank you.

Moreover, the addition of APP can improve the tensile strength, elongation at break and elastic modulus of the film (Food Hydrocolloid., 2014, 35, 287-296.). There is no study on the preparation of active film by adding APP into SA. Yuan et. al. proposed that with the increase of tea polyphenol content, the tensile strength, fracture strain and water vapor transmission rate of the film increase. Moreover, tea polyphenols can enhance its inflammatory properties and promote wound healing. The composite film with polyphenols can be used not only as food packaging, but also as wound dressing (Food Hydrocolloid., 2019, 97, 105197.). The above research fully shows that adding APP into SA to prepare active packaging materials and its application in food packaging has research significance.

3. Lines 134-135 – Modifying the formula for Swelling capacity or swelling ratio is not enough, the graphical representations also had to be modified.

Re: I would like to reiterate that the swelling test method used in this experiment is indeed the same as that described in the neutral energy test section of the present paper, and the calculation method is also to calculate the percentage of mass difference before and after immersion. The data and method are correct. I can show you our calculation process, for example, our SA/APP composite membrane M1 is 0.0240, M2 is 0.0606, after calculation, its swelling rate is 152.5%. Indeed, I am very sorry for the wrong description of swelling rate, but our data and test method are true and correct, I can be responsible for this. The data in the article are calculated by careful and scientific methods. Moreover, you can see from other people's studies that the swelling rate of sodium alginate composite membrane containing glycerol is 157.38%, and that of calcium chloride crosslinked membrane is 60% - 300%, which fully shows that our parameters are normal and correct (Materialia, 2020, 12, 100827.). The title of the article is Evaluation of diclofenac sodium incorporation in alginate membranes as potential drug release system. I don't know how to modify the correct data. Thank you.

4. Lines 222-223, it is not specified how the strawberries were covered with the obtained film: „Strawberries were directly coated with the film of the experimental film and the control film”??, the fruits were directly coated or were kept in trays made of another material and then covered with film? Please specify this. If the fruits has been covered, specify how to carry out the coating and drying process. If they were packaged in trays, then the antioxidant activity of APP is not relevant.

Re: Strawberries are directly wrapped with prepared films. There is no way to put strawberries into the tray or coating them by film solution. Because the films have been prepared and dried before the preservation experiment, the strawberries can be wrapped directly with the film, which is the same method used in the study of Lan et al (LWT, 2019, 113, 108297.). I have added this description in section 2.11, because 2.11 is the summary of preservation experiment, so I will not repeat it in other performance test sections. Thank you.

In the experiment, the 8-point, uniformly-colored strawberry (Fragaria × ananassa Duch. cv. Benihoppe), and the post-harvest strawberries were pre-cooled and randomly grouped. After weighing and recording all strawberries, all strawberries were directly wrapped with SA/AgNPs/APP, SA/APP and SA films which were made before by electrospray. The strawberries without film were used as air control group and commercially available PE film wrapped strawberries as a PE control group. The strawberries of all groups were stored in the refrigerator with 75% humidity and 4 temperature. Take out strawberries every 2 days, remove the external film for weighing, photographing, sensory evaluation and hardness measurement, and then grind the strawberries with a mortar to test other indicators, a total of 12 days, each test is conducted 5 times.

2.11. Weightlessness, sensory, and decay

After taking out the strawberries, directly tear off the film wrapped on the surface for weighing.

5. Line 460, figure 6, Why using a temperature range from 50 to 350°C in thermal analysis if you are interested only in the temperature range in which they are stored the strawberries?.

Re: I'd be happy to explain it again for you. First of all, we test the DSC performance to test the thermal stability of the composite film, because the polymer such as sodium alginate will have water evaporation, depolymerization and thermal decomposition at a certain temperature. The substrate of the composite film is sodium alginate. However, when we add apple polyphenols and nano silver, there will be interaction between sodium alginate and apple polyphenols, which will cause the change of its thermal stability, which is reflected in the change of temperature point and the change of energy required. We can see from the change of temperature that the composite film is not a simple physical mixing, but there is interaction between the three (hydrogen bond). We can also see at what temperature the composite film denaturates, which is a reference for its loss of commercial value at a certain temperature. The purpose of DSC is to characterize the materials, which has little relationship with the application objects. We can't measure the properties of the composite film by the application object.

Secondly, I don't know if your doubt is that 4 is not included in the temperature range of DSC (50-350), I need to make it clear that the starting temperature of DSC test is 20. You can see from other people's research that the starting temperature of DSC is also 20. There is no scientific problem (Int. J. Biol. Macromol., 2020, 145, 124-132.).

Moreover, it can be seen from the DSC diagram that the low temperature is the process of water evaporation, which has no guiding significance for the thermal stability of materials. Generally speaking, low temperature is a drying process, and there will be no chemical reaction. As for 4, the difference between high or low moisture content of the film itself will not affect the thermal stability of the material. What's more, you may wonder, is the application environment not important to the stability of the material? The answer is very important, but we should consider whether this temperature is meaningful to the thermal stability of materials. If a material needs to be tested at 4 for its thermal stability and structural change, it indicates that the material itself has unique properties or is not suitable for the preparation of packaging materials. It can not stably exist at room temperature and is not suitable for any packaging. The composite film does not belong to any of the above categories. If the film is applied to strawberries at room temperature of 25, should DSC test temperature be 4 or 25? I don't think it's a shift of will.

In some literatures, the starting temperature of DSC is 350. It can be seen that the properties of polymers are the main factors affecting the DSC test range, followed by effective and meaningful application environment conditions (Vacuum, 2009, 83, S182-S185.).

6. Lines 583-586, Discussions about polyphenols extracted from guava leaves that have a different composition and cannot be compared with apple polyphenols, and that polyphenol mixtures are usually much more active than any single polyphenol alone [37]”. ??

Re: Previous studies have suggested that the combinations of rutin and quercetin, quercetin andquercitrin, kaempherol and rutin showed better antibacterial activity than either flavonoid alone (Biosci. Biotech. Bioch., 2002, 66, 1009-1014.). There is a study to prove this conclusion. “Arima, Hidetosi, Ashida, et al. Rutin-enhanced Antibacterial Activities of Flavonoids against Bacillus cereus and Salmonella enteritidis (Food & Nutrition Science)[J]. Bioscience Biotechnology & Biochemistry, 2002, 66, 1009-1014.” Thank you.

7. Lines 614-618, they are not necessary and do not answer the subject.

Re: It is my mistake to express that app has a good effect on litchi preservation, which shows that the application of SA/AgNPs/APP film in fruit preservation has a development prospect. I made a change to this, thank you.

The application of APP in litchi preservation can significantly inhibit the browning of litchi, enhance the activity of antioxidant enzymes, and improve the shelf life of litchi (Int. J. Biol. Macromol., 2018, 114, 545-555). Phenolic compounds with biological activity can play a role in physiological changes of microbial cell membrane and eventually lead to bacterial death. The results showed that APP had an important effect on inhibition of fruit surface microbial activity and browning. The application of SA/AgNPs/APP film in fruit preservation has a development prospect (Int. J. Biol. Macromol., 2018, 114, 545-555).

8. Lines 691-698, Comparisons are made with other results obtained in other specialized works, but they do not refer to the same methods or treatments used, to the same types of film or film-coated products or the same fruits.

Re: Added, thanks. The reaserch of Kanikireddy et al. showed the same trend. During storage, the weight loss rate and decay rate of unpacked strawberries were higher than those packed with carboxymethyl cellulose/guar gum/AgNPs composite film, and the water moved to the surrounding environment faster. Because of its antioxidant effect of mint leaf extract and antibacterial effect of AgNPs, the decay and ripening of strawberry were inhibited (Carbohyd. Polym., 2020, 236, 116053.). Similarly, after strawberry was stored at 4℃ for 3 days, mold appeared on the unpacked samples. The plyvinyl alcohol composite film added with tea polyphenols could effectively prolong the shelf life of strawberry by 3 days and effectively inhibit the biological activity of mold (LWT., 2019, 113, 108297.). Due to the excellent antibacterial property of AgNPs, the preservation effect of SA/AgNPs/APP composite film on strawberry was better. SA has a high barrier, which can effectively inhibit the transpiration and respiration of fruits, which has a positive effect on prolonging the shelf life of fruits. Therefore, the composite film is a commercial application to maintain the weight of fruits in cold storage (Food Chem., 2019, 270, 385-394.).

Round 3

Reviewer 2 Report

The authors answered only two of the nine requirements. I do not agree with the publication of the paper.

Author Response

Comments and Suggestions for Authors

The authors answered only two of the nine requirements. I do not agree with the publication of the paper.

Re: I'm so sorry. All the questions I received last time are as follows. I will answer each question in detail. I will revise those questions carefully, thank you.

  1. Lines 58-64, A comparison with chitosan is not necessary, as it is a good antimicrobial agent.

Re: This passage is not introduced here to compare the antibacterial properties of APP and chitosan. It is because the research results show that the antibacterial and antioxidant properties of APP are enhanced when APP is added into the chitosan composite film, which indicates that APP has antibacterial and antioxidant properties. Because of my mistakes in words, this paragraph is easy to be misunderstood, so I modified this part. Thank you.

According to this, APP has been added into the active packaging materials in recent years to prepare antibacterial and antioxidant active packaging with good results. Sun et al. (Carbohyd. Polym., 2017, 163, 81-91.) mixed APP into chitosan to develop a new functional film. The results showed that adding APP to chitosan considerably improved the physical properties of the film, and increased the antioxidant and antibacterial activities. Moreover, the addition of APP can improve the tensile strength, elongation at break and elastic modulus of the film (Food Hydrocolloid., 2014, 35, 287-296.). There is no study on the preparation of active film by adding APP into SA. Yuan et al. proposed that with the increase of tea polyphenol content, the tensile strength, fracture strain and water vapor transmission rate of the film increase. Moreover, tea polyphenols can enhance its inflammatory properties and promote wound healing. The composite film with polyphenols can be used not only as food packaging, but also as wound dressing (Food Hydrocolloid., 2019, 97, 105197.). The above research fully shows that adding APP into SA to prepare active packaging materials and its application in food packaging has research significance.

  1. Lines 60-61, It compares the films have different base compositions.

Re: The substrate of this study is chitosan, which is different from sodium alginate and should not be used for comparison. But I don't want to compare with it here, through this study, I explain the antibacterial, antioxidant and mechanical properties of APP. I'm so sorry for the misunderstanding. I have rewritten this part and added other examples to illustrate my intention. Thank you.

According to this, APP has been added into the active packaging materials in recent years to prepare antibacterial and antioxidant active packaging with good results. Sun et al. (Carbohyd. Polym., 2017, 163, 81-91.) mixed APP into chitosan to develop a new functional film. The results showed that adding APP to chitosan considerably improved the physical properties of the film, and increased the antioxidant and antibacterial activities. Moreover, the addition of APP can improve the tensile strength, elongation at break and elastic modulus of the film (Food Hydrocolloid., 2014, 35, 287-296.). There is no study on the preparation of active film by adding APP into SA. Yuan et al. proposed that with the increase of tea polyphenol content, the tensile strength, fracture strain and water vapor transmission rate of the film increase. Moreover, tea polyphenols can enhance its inflammatory properties and promote wound healing. The composite film with polyphenols can be used not only as food packaging, but also as wound dressing (Food Hydrocolloid., 2019, 97, 105197.). The above research fully shows that adding APP into SA to prepare active packaging materials and its application in food packaging has research significance.

  1. Lines 134-135 – Modifying the formula for Swelling capacity or swelling ratio is not enough, the graphical representations also had to be modified.

Re: There are differences between us on this issue. I also believe that you are a very professional teacher. So, I did a new experiment on the swelling rate. According to the new experimental results, it prove that your previous opinion is correct. I am so sorry for your previous opinions. I have analyzed the data and the analysis part. Thank you very much for your professional opinions. Thank you very much.

The effect of the ratio of SA with AgNPs and ultrasonic treatment time in the swelling-ratio  behavior properties of SA/AgNPs/APP films was evaluated, and the results are shown in Figure 2c. SR of SA/AgNPs/APP films were 60.90% (10:0), 45.85% (9:1), 38.29% (8:2) and 36.23% (6:4), respectively, showing the high water hydrophilic characteristic. The higher the concentration of SA, the higher the swelling degree of the film preparation, which can also be observed in the SA/AgNPs/APP film (Carbohyd. Polym., 2020, 235, 115747.) After added AgNPs, the minimum SR of the SA/AgNPs/APP films was reached to 28.37% (decreased by 32.63%) in 7:3. The high values of SR of the SA/APP (10:0) films may be attributed to the strong hydrophilicity of SA and APP that can easily interact with water molecules (Carbohyd. Polym., 2017, 163, 81-91.). The decrease in SR indicated water hydrophobic of the SA/APP films were enhanced for the hydrophobic characterization of AgNPs. These differences in the SR changes could be due to the SA/APP content used in each formulation. First, increasing the SA/APP concentration results in a higher number of reactive sites to interact with water, thus increasing the water absorption. On the contrary, it shows why the concentration of AgNPs increases while the SR value decreases. The number of -OH in SA/APP film decreased after incorporation of AgNPs, thus it could combine less water [35]. In addition, this may be due to the interaction between SA polymer chain and AgNPs, which limits the movement chain of SA polymer (Food Chem., 2019, 283, 397-403.). It is worth noting that ultrasonic treatment can significantly reduce the SR of the composite film. When it is 7:3-30 min, the lowest SR is 7.52%. Due to the special behavior of the grain boundary, the hydrophobic repulsion of AgNPs is caused, while the ultrasonic wave disperses AgNPs uniformly in the matrix for a certain time, which hinders the swelling caused by water molecules. Secondly, the uniform AgNPs has certain resistance to the invasion of water molecules, which eventually leads to the decrease of SR. However, the ultrasonic time transition leads to the aggregation of AgNPs and the formation of pores, which results in the increase of SR (Mater. Chem. Phys., 2019, 235, 121760.). As for the decrease of 10:0 SR caused by ultrasound, it may be due to the more closed matrix caused by ultrasound, which makes the blend film more difficult to contact water, and the surface crosslinking degree of CaCl2 is high (Ultrason. Sonochem., 2017, 36, 11-19.). Su-Giz et al. pointed out that the limit SR of the crosslinked SA film in water is between 50% – 70%, and the SR of the SA/AgNPs/APP film is within the normal range. The prepared composite film can rapidly swell in water for 5 minutes, which is very useful for the release of active substances. The SA composite film is suitable for wound dressing and food packaging ( Int. J. Biol. Macromol., 2020, 148, 49-55.). The swelling percentage of sodium alginate composite film in acid buffer solution is up to 95%, and it has great swelling characteristics under acidic conditions. When the composite film is applied to strawberry preservation research, the rotten juice of strawberry can stimulate the film swelling and release antibacterial and antioxidant substances, which brings advantages to the system (Carbohyd. Polym., 2020, 235, 115747.).

  1. Line 460, figure 6, Why using a temperature range from 50 to 350°C in thermal analysis if you are interested only in the temperature range in which they are stored the strawberries?.

Re: I'm sorry, DSC is to determine its thermal stability, so we refer to a large number of literatures and find that sodium alginate belongs to the stage of free water evaporation when the temperature is lower than 100 . Therefore, we choose to record the temperature rising from room temperature (25 ) according to most literatures (Chem. Eng. Sci., 2020, 3, 116101.).

  1. Lines 222-223, it is not specified how the strawberries were covered with the obtained film: „Strawberries were directly coated with the film of the experimental film and the control film”??, the fruits were directly coated or were kept in trays made of another material and then covered with film? Please specify this. If the fruits has been covered, specify how to carry out the coating and drying process. If they were packaged in trays, then the antioxidant activity of APP is not relevant.

Re: Strawberries were directly coated with the film of the experimental film and the control film that we have prepared before. I have revised and supplemented this part.

In the experiment, the 8-point, uniformly-colored strawberry (Fragaria × ananassa Duch. cv. Benihoppe), and the post-harvest strawberries were pre-cooled and randomly grouped. After weighing and recording all strawberries, all strawberries were directly wrapped with SA/AgNPs/APP, SA/APP and SA films which were made before by electrospray. The strawberries without film were used as air control group and commercially available PE film wrapped strawberries as a PE control group. The strawberries of all groups were stored in the refrigerator with 75% humidity and 4 temperature. Take out strawberries every 2 days, remove the external film for weighing, photographing, sensory evaluation and hardness measurement, and then grind the strawberries with a mortar to test other indicators, a total of 12 days, each test is conducted 5 times.

2.11. Weightlessness, sensory, and decay

After taking out the strawberries, directly tear off the film wrapped on the surface for weighing. The weightlessness was tested with an analytical balance (FA124, Shanghai Meiyingpu Instrument Manufacturing Co., Ltd., China), which was estimated as the percentage of initial weight loss. (LWT, 2019, 113, 108297.) 

  1. Lines 583-586, Discussions about polyphenols extracted from guava leaves that have a different composition and cannot be compared with apple polyphenols, and that polyphenol mixtures are usually much more active than any single polyphenol alone [37]”. ??

Re: It has been discussed and added. Thank you. Luo et al. showed that guava leaf extract had antibacterial effect on E. coli and S. aureus, and the higher the concentration, the more obvious the antibacterial effect. And they suggest that different types of polyphenols have different effects on bacteria, and that polyphenol mixtures are usually much more active than any single polyphenol alone (Materials, 2019, 12, 2923-2928.). This may be due to the different sensitivity of microorganisms to different polyphenols. The sensitivity of bacteria to polyphenols depends on the species of bacteria and the structure of polyphenols (Food Chem., 2019, 284, 108-117.). Catechins have direct antibacterial effect by destroying bacterial cell membrane, inhibiting fatty acid synthesis and inhibiting enzyme activity (Carbohyd. Polym., 2018, 196, 162-167.). Plant extract is a complex mixture of many active substances acting simultaneously. Therefore, the antibacterial and antioxidant properties may play an accumulative role in its action (Carbohyd. Polym., 2018, 196, 162-167.). This conclusion was also put forward in Olszewska et al 's study (Food Res. Int., 2020, 134, 109214.). In addition, biological activity may interfere with the results. Therefore, further study on the model system of plant extracts may be very helpful to study the possible synergistic or antagonistic effects.

  1. Lines 614-618, they are not necessary and do not answer the subject.

Re: It is my mistake to express that APP has a good effect on litchi preservation, which shows that the application of SA/AgNPs/APP film in fruit preservation has a development prospect. I made a change to this, thank you.

In addition, the overall antioxidant capacity of SA/AgNPs/APP film may also be the result of the synergistic action of different active substances including phenolic acids, flavonols, flavonoid-3-ol, anthocyanins, dihydrochalcone and procyanidins in the matrix rather than single compounds. Plant extracts are a complex mixture of many antioxidants that act simultaneously. Therefore, antioxidants may have cumulative effects, but synergistic and antagonistic effects are also possible. In addition, there may be interference between antioxidants and biomaterials (Food Chem., 2019, 284, 108-117.). The antioxidant activity is closely related to the purity and source of the classification, and the antioxidant activity is also closely related to its antibacterial property (J. Sci. Food Agr., 2012, 92(15), 2983-2993.). The application of APP in litchi preservation can significantly inhibit the browning of litchi, enhance the activity of antioxidant enzymes, and improve the shelf life of litchi (Int. J. Biol. Macromol., 2018, 114, 547-555.). Phenolic compounds with biological activity can play a role in physiological changes of microbial cell membrane and eventually lead to bacterial death. The results showed that APP had an important effect on inhibition of fruit surface microbial activity and browning. The application of the film in fruit preservation has a development prospect (Carbohyd. Polym., 2017, 163, 81-91.). Therefore, it is of great significance to apply the SA/AgNPs/APP composite film containing APP in food, such as strawberry preservation and delaying strawberry aging.

  1. Lines 691-698, Comparisons are made with other results obtained in other specialized works, but they do not refer to the same methods or treatments used, to the same types of film or film-coated products or the same fruits.

Re: I misunderstood your meaning before. Now I compare the performance of each preservation. You can see it in lines 733, 741, 774, 826, 852, 895, 915 and 926. Thank you
